# Geophysical constraints on the properties of a subglacial lake in northwest Greenland

Ross Maguire[1,2], Nicholas Schmerr[1], Erin Pettit[3], Kiya Riverman[4], Christyna Gardner[5], Daniella N. DellaGiustina[6], Brad Avenson[7], Natalie Wagner[8], Angela G. Marusiak[1], Namrah Habib[6], Juliette I. Broadbeck[6], Veronica J. Bray[6], Samuel H. Bailey[6]

[1]Department Geological Sciences, University of Maryland, College Park MD, 20742, USA
[2]Department of Earth and Planetary Sciences, University of New Mexico, Albuquerque NM, 87131, USA
[3]College of Earth, Ocean, and Atmospheric Sciences, Oregon State University, Corvallis OR, 97331-5503, USA
[4]Courant Institute of Mathematical Sciences, New York University, New York NY 10012 USA
[5]Department of Geosciences, Utah State University, Logan UT, 84322-4505, USA
[6]Lunar and Planetary Laboratory, University of Arizona, Tucson AZ, 85721-0092, USA
[7]Silicon Audio Inc., Austin TX, USA
[8]Department of Geosciences, University of Alaska, Fairbanks AK, 99775, USA

*Correspondence to*: Ross Maguire (rmaguire@umd.edu)

**Abstract.**

In this study, we report the results of an active source seismology and ground-penetrating radar survey performed in northwestern Greenland at a site where the presence of a subglacial lake beneath the accumulation area has previously been proposed. Both seismic and radar results show a flat reflector approximately 830 - 845 m below the surface, with a seismic reflection coefficient of -0.43 +/- 0.17, which is consistent with the acoustic impedance contrast between a layer of water below glacial ice. Additionally, in the seismic data we observe an intermittent lake bottom reflection arriving between 14 - 20 ms after the lake top reflection, corresponding to a lake depth of approximately 10 - 15 m. A strong coda following the lake top and lake bottom reflections is consistent with a package of lake bottom sediments although its thickness and material properties are uncertain. Finally, we use these results to conduct a first-order assessment of the lake origins using a one-dimensional thermal model and hydropotential modeling based on published surface and bed topography. Using these analyses, we narrow the lake origin hypotheses to either anomalously high geothermal flux or hypersalinity due to local ancient evaporite. Because the origins are still unclear, this site provides an intriguing opportunity for the first *in situ* sampling of a subglacial lake in Greenland, which could better constrain mechanisms of subglacial lake formation, evolution, and relative importance to glacial hydrology.

**1 Introduction**

There is mounting evidence that subglacial lake systems below the Antarctic and Greenland ice sheets play an important role in glacier dynamics and ice-sheet mass balance considerations. In Antarctica, the presence of subglacial lakes is suspected to promote ice flow by reducing basal shear stress (e.g., Bell et al., 2007), and periodic drainage events have been linked to accelerated ice flow in outlet glaciers and ice streams (e.g., Stearns et al., 2008; Siegfried et al., 2016). Similarly, in Greenland subglacial lake systems also provide a reservoir for the storage of surface or basal melt water, and hence may be an important, but largely unknown, factor in global sea level change. Additionally, subglacial lakes are of interest due to their ability to harbor complex microorganisms adapted to extreme environments (Achberger et al., 2016; Campen et al., 2019; Vick-Majors et al., 2016) and for paleoenvironmental information contained in subglacial lake sediments (Bentley et al., 2011).

While the presence and nature of subglacial lakes underlying the Antarctic ice sheet has been studied for more than 50 years, the existence of subglacial lakes below the Greenland ice sheet is a relatively recent discovery and comparatively little is known about their properties and origin. Detection of subglacial lakes has relied on a variety of methods, including radio-echo sounding (Robin et al., 1970; Siegert et al., 1996; Langley et al., 2011; Palmer et al., 2013; Young et al., 2016; Bowling et al., 2019) satellite altimetry measurements (Fricker et al., 2007; Palmer et al., 2015; Siegfried & Fricker, 2018; Willis et al., 2015), and active source seismic experiments (e.g., Horgan et al., 2012; Peters et al., 2008). Using these techniques, approximately 400 subglacial lakes have been detected in Antarctica (Wright & Siegert, 2012), of which 124 are considered "active" by Smith et al., (2009). In Greenland, subglacial lakes were first detected in radio-echo sounding data by Palmer et al. (2013), who identified two small (roughly 10 km$^2$) flat regions of anomalously high basal reflectivity below the northwestern Greenland ice sheet. These features, named "L1" and "L2", were discovered below 757 m and 809 m of ice respectively. Recently, Bowling et al. (2019) greatly expanded the inventory of subglacial lakes in Greenland to approximately 54 candidates based on a combination of airborne radio-echo sounding and satellite altimetry data. The new inventory shows that, in contrast to subglacial lakes in Antarctica which tend to form under thick (> 4 km) warm-based ice in the continental interior, the majority of subglacial lakes in Greenland are found under relatively thin (1 - 2 km) ice near the margins of the ice sheet. Bowling et al. (2019) find that most subglacial lakes in Greenland appear to be stable features, showing temporally consistent radio-echo sounding signatures and an absence of vertical surface deformation over the decadal time scales of observation. Of the 54 candidate lakes, only 2 showed signs of vertical surface deformation indicative of active draining or recharge.

The formation and location of the detected subglacial lake features in Greenland remains elusive because many are located in regions where observations and modeling suggest that the base of the ice is frozen to its bed (MacGregor et al., 2016). Complicating our understanding of the nature of subglacial lakes is the fact that uniquely identifying lakes in radar data is challenging since basal reflectivity is sensitive to both the physical properties and the roughness of the material underlying the ice (e.g., Jordan et al., 2017). Amplitude anomalies of radar echoes in the range of +10 to +20 dB are often interpreted as

subglacial lakes, although flat regions of saturated sediment may produce similar anomalies. Furthermore, the total volume of water stored in subglacial lake systems is unknown since airborne and space based remote sensing observations are incapable of measuring lake depth (i.e., water column thickness).

Seismic investigations provide an independent means of confirming the presence of subglacial lakes and are capable of measuring lake depth and underlying geological structures which can provide valuable clues into their formation and total volume. For example, Peters et al. (2008) performed an active source seismic survey near the South Polar region of Antarctica, and observed reflections from both the top and bottom of a subglacial lake that lies 2.8 km below the ice surface, which allowed them to image a lake depth of about 32 m and infer the underlying sedimentary structure. Additionally, Woodward et al., (2010) performed an active source seismic investigation of lake Ellsworth in west Antarctica, which lies at the bottom of a narrow subglacial valley below approximately 3 km of ice. They found large variations in lake depth from between 52 m to 156 m and were able to estimate the total volume of liquid water to be 1.37 $km^3$. Later, Smith et al. (2018) reanalyzed the data to investigate the sedimentary structure below lake Ellsworth, and found evidence of a thin sedimentary package (minimum thickness of 6 m), which they suggest may have built up slowly over at least 150 ka. This contrasts to results from seismic investigations of Lake Vostok, the largest of Antarctica's subglacial lakes, which show evidence for a much thicker water column (up to 1100 m) and a thicker layer of lake bottom sediments (up to 400 m) below approximately 4 km of ice (e.g., Filina et al., 2008). Seismic investigations have also been useful for illuminating the properties of subglacial lakes below much thinner ice columns in active ice streams, such as subglacial Lake Whillans which is situated below approximately 800 m of ice and has a maximum water column thickness of less than 10 m (e.g., Horgan et al., 2012).

## 2 Methods

### 2.1 Field experiment

In June 2018, we conducted a geophysical survey in northwestern Greenland above the candidate subglacial lake feature named "L2" by Palmer et al. (2013). This feature sits within a 980 $km^2$ drainage basin, is roughly adjacent (< 10 km) to the nearest ice divide (Fig. 1a and 1b), and within an accumulation area. Using RACMO2 1-km resolution modeling of Greenland's near surface climate and surface mass balance (Noël et al., 2018), we estimate the mean annual air temperature to be -22° C. This model is forced with ERA-Interim reanalysis climate information (Dee et al., 2011) at the boundaries and evaluated with in situ observations. The mean annual snow accumulation rate at the field site is ~0.3 m $yr^{-1}$ ice equivalent. In order to confirm the presence of the subglacial lake and investigate its physical properties, we collected data using both active source seismology and ground-penetrating radio-echo sounding (GPR).

The active source seismic experiment (Fig. 1c) consisted of a moving line of 24 40-Hz vertical component geophones spaced 5 m apart. For each line, we collected data at 4 shot locations using an 8 kg sledgehammer impacted against a 1.5 cm thick steel plate. At each source location at least 5 hammer shots were stacked into a single shot gather in order to increase the signal to noise ratio. The first shot location of each line was offset 115 m from the first geophone, and subsequent shot locations were moved 115 m along the line. After data were collected for each of the 4 shot locations, the line was moved 230 m east along the traverse and data collection was repeated. The seismic line was moved a total of 10 times, totalling 40 separate shot locations. Using this geometry, we obtained reflection points at the ice bottom spaced every 2.5 m along a traverse totalling 2400 m (Fig. 1d). We created a seismic reflection image by bandpass filtering data between 100 - 200 Hz and applying a normal moveout (NMO) correction with a velocity of 3700 m s$^{-1}$, which was found to be the average velocity of the ice column from NMO analysis of the primary bed reflection. High frequency spatial noise with wavenumber greater than 0.05 m$^{-1}$ was removed with f-k filtering. Shot gathers with offsets of −115 m and 230 m from the first geophone contained an air wave arrival that was muted by zeroing a 10 ms window with a moveout of 315 m s$^{-1}$.

The GPR data was collected across a ~5.5 km transect roughly parallel to the seismic survey (Fig. 1c), using an acquisition system specially adapted to be towed by a motor sled traveling at approximately 10 km h$^{-1}$ (e.g., Welch & Jacobel, 2003). The system used a Kentech pulse transmitter that produces +/- 2000 V pulses with a variable pulse repetition frequency of between 1 and 5 kHz. The antennae are resistively loaded wire dipoles with nominal frequency of 5 MHz, and the receiver uses an 8-bit NI USB-5133 digitizer and a computer. We stacked 64 traces over 10 - 15 m horizontal distance and then we filtered between 2 – 8 MHz in post processing to produce each final trace on the radargram. We created a GPR reflection image by converting the radar data to depth using a radar velocity of 172 m μs$^{-1}$ (see Supporting Information).

## 2.2 Basal radar reflectivity

We estimated the relative basal reflectivity of the bed reflector along the track by first correcting for geometric spreading, then correcting for englacial attenuation assuming the englacial attenuation rate is uniform. This assumption of uniform englacial attenuation is common (e.g., Christianson et al., 2016; Palmer et al., 2013), but not ideal for this situation because horizontal variability in the thermal structure of the ice is not well constrained. We picked the peak power along our bed profile using a semi-automated picking routine, where the user provides the approximate bed picks to guide the automated routine. We assume an englacial average attenuation rate of -15 dB km$^{-1}$ which is the lower end on the range of values suggested for northwest Greenland by MacGregor et al. (2015), which are based on tracing the return power of reflections from internal ice layers (e.g., Matsuoka et al., 2010). We chose the lower end based on fitting a linear curve to peak power versus depth for our data set, which suggests attenuation between -12 dB km$^{-1}$ and -20 dB km$^{-1}$. This method, described by Jacobel et al. (2009) and further assessed and compared to other methods by Hills et al. (2020), has limitations for our data set because of the 1) limited depth range, 2) limited spatial sampling, 3) scatter in the data due to noise, 4) it relies on the assumption of uniform horizontal

attenuation, and 5) it only applies to the depth range of our data; therefore, we only use this estimate as rough proxy for basal material. Because of uncertainties in the attenuation assumptions, we also provide the correction factors for -25 dB km[-1] attenuation.

### 2.3 Basal seismic reflectivity

We calculate the reflection coefficient at the base of the ice by analyzing the amplitudes of the primary bed reflection and its multiple, which we refer to as $R1$ and $R2$ from hereon. When both $R1$ and $R2$ are visible, the basal reflection coefficient $c_R$ can be determined as a function of incidence angle $\theta$ using Eq. (1) where $A_{R1}$ and $A_{R2}$ are the amplitude of the first and second ice bottom reflections, respectively, $a$ is the absorption coefficient, and $L$ is the raypath length of the $R1$ reflection (e.g., Peters et al., 2008).

$$c_R(\theta) = 2\frac{A_{R2}(\theta)}{A_{R1}(\theta)}\, e^{aL(\theta)}$$

(1)

At a given geophone, two factors control the amplitude ratio between $R1$ and $R2$. First, $R1$ and $R2$ reflect off of the lake with slightly different angles, which changes the relative amount of energy partitioned into each reflection. Second, since $R2$ travels farther than $R1$, its amplitude is diminished due to geometrical spreading and attenuation. However, at incidence angles in this study, the difference in reflection coefficients between $R1$ and $R2$ is negligible. Additionally, the path lengths of $R1$ and $R2$ vary by < 5% between their shortest and farthest offsets. Therefore, to calculate the reflection coefficient $c_R$ we use the normal
incidence approximation and compare amplitude ratios $A_{R2}/A_{R1}$ on individual seismograms. In order to minimize the influence of the air wave on $A_{R2}/A_{R1}$ ratio we exclude data from geophones with offsets between 135 – 155 m, where there is potential interference between $R1$ and the air wave. Measurements of $A_{R1}$ and $A_{R2}$ are made prior to f-k filtering.

    The relationship between the absorption coefficient $a$ and the seismic quality factor $Q$ is given by Eq. (2), where $c$ is the seismic
velocity, and $f$ is frequency (Bentley & Kohnen, 1976). While in principle, the spectral ratio of the $R1$ and $R2$ reflections can be used to determine the attenuation ($Q^{-1}$) of the glacial ice (Dasgupta & Clark, 1998; Peters et al., 2012) the low signal to noise ratio of the $R2$ reflection prevents us from making a robust measurement. Here, we estimate the absorption coefficient $a$ based on the study of Peters et al. (2012), who reported $Q = 355$ +/- 75 in the upper 1 km of ice in Jakobshavn Isbrae, western Greenland. Using Eq. (2) with $c = 3.7$ km s[-1], and assuming a frequency of 100 Hz (the predominant frequency observed in
the reflections), this corresponds to an absorption factor $a = 0.23$ +/- 0.06 km[-1].

$$Q^{-1} = \frac{ca}{\pi f}$$

(2)

## 3 Results

The seismic reflection profile (Fig. 2a) shows a clear ice bottom reflection (R1) across the entire transect arriving with a two-way travel time between 400 – 460 ms. The ice bottom multiple $R2$ is also visible between 800 – 920 ms. At transect distances between 0 – 1700 m, the $R1$ reflection is flat and relatively uniform in character, which we interpret to be the signal of the top of the subglacial lake. In this region, $R1$ arrives at 457 ms, which corresponds to a depth of 845 m, assuming an average $V_P$ of 3700 m s$^{-1}$ within the ice. At larger transect distances, the reflections arrive earlier with increasing distance, which likely reflects the bed topography adjacent to the subglacial lake. An additional reflection is observed arriving between 14 – 20 ms after $R1$, which we interpret as a lake bottom reflection (Fig. 2b). This signal is intermittently observed but is most continuous at transect distances between 660 – 1200 m. The travel time differential between the lake top and lake bottom reflection is used to measure the thickness of the water column as a function of distance along the transect. Assuming $V_P$ in the lake of 1498 m/s (Table 1), the lake is between 10 – 15 m deep (Fig. 2c). An uncertainty of +/- 50 m s$^{-1}$ on the seismic velocity of the lake would correspond to a lake depth uncertainty of +/- 0.5 m. A strong coda following the lake bottom reflection is apparent which is likely caused by a thin (~ 10 m) sediment package underlying the lake (see Discussion).

In the GPR profile, the subglacial lake is apparent as a flat reflector at an elevation of ~510 m along the majority of the transect (Fig. 3a). The surface topography slopes gently to the west across the transect, hence the lake top is slightly deeper (i.e., the ice is thicker) towards the east (Fig. 3b). The lake is beneath 840 m of ice at transect distances between 2 to 4.5 km, which roughly corresponds to the location of the seismic survey. The transition from the lake top to the adjacent bed is observed at approximately 4100 m along the transect. In addition, we observe that the bed reflected power is approximately 5 dB higher over the lake compared to the surrounding region (Fig. 3c). Similar to the conclusion of Palmer et al., (2013), which was based on airborne radar, we infer this elevated reflectivity to result from an ice/water interface. However, Tulaczyk & Foley (2020) show that subglacial materials with high conductivity can produce similar reflections to an ice/water interface. Additionally, Tulaczyk & Foley (2020) provide a method using information about phase and multiple frequencies to better distinguish among freshwater, brine, or water- or brine-saturated clay. Our available data, however, are at a single frequency and do not retain phase information; therefore, we do not have sufficient information to distinguish between these high conductivity materials based on radar alone. The secondary seismic reflection discussed above suggests that the lake is water of unknown salinity, rather than saturated sediments.

Assuming an absorption factor of $a = 0.23$, the average seismic reflection coefficient of the lake bottom across the transect is -0.43 +/- 0.17 (Fig. 4a). In Fig. 4b, we plot $c_R$ calculated for each shot gather above the lake as a function of the distance along the transect. For comparison we show the expected reflection coefficients of several different geologic materials underlying glacial ice. Beyond the boundary of the lake, the $R2$ signal strength is diminished and we are unable to confidently measure

$c_R$. The reflection coefficients were modeled using the two-term approximation of the Zoeppritz equations (e.g., Aki & Richards, 2002; Booth et al., 2015) with the material properties shown in Table 1. In contrast to other likely geological materials at the base of the ice, liquid water is expected to have a negative reflection coefficient. The reflection coefficient modeled for lithified sediments or bedrock underlying ice is similar in amplitude to liquid water but opposite in sign, thus, without polarity information sedimentary rock strata could be mistaken for a lake signature. Here, we measure $R1$ with an opposite polarity of the source (see Fig. S4), thus, liquid water is the most likely explanation. However, if we are significantly overestimating the magnitude of reflection coefficient, due to, for example, the large uncertainties on the attenuation structure of the ice, a layer of water saturated dilatant till may also be able to explain our data.

## 3 Discussion

### 3.1 Lake geometry and volume

If our interpretation of the observed seismic and radar reflections as signals from the lake top and bottom is correct, it implies that L2 could hold a significant volume of water. Assuming the imaged lake depth of approximately 15 m is representative of average lake depth throughout the roughly 10 km$^2$ surface area determined by radio-echo sounding, we estimate the total volume of liquid water to be 0.15 km$^3$ (0.15 Gt of water). While this is only a small fraction of the 217 +/- 32 Gt of ice that Greenland is estimated to lose each year to glacier discharge and surface melting (Shepherd et al., 2019), the net storage capacity of all of Greenland's subglacial lakes could be appreciable.

To verify our interpretation of the lake top and bottom seismic reflections, we modeled synthetic seismic waveforms of the 12[th] shot gather in our survey, which contained some of the clearest reflections. This shot gather corresponds to transect distances between 660 - 720 m in the seismic reflection image. Synthetic seismograms were computed using Specfem2D (Tromp et al., 2008) for two simple layered models of a 12 m thick lake underlying 850 m of glacial ice. In the first model the lake is underlain by a thick layer of sediments that extends to the bottom of the model domain. In the second model there is 10 m of sediments overlying a discontinuity with the bedrock below. The seismic velocity profiles for the two cases are shown in the insets in Fig. 5b and 5c. The source used in the simulations was a Ricker wavelet with a dominant frequency of 100 Hz. Fig. 5 shows a comparison between the observations and synthetics. In both the observed (Fig. 5a) and synthetic (Fig. 5b and 5c) shot gathers, the lake top and lake bottom reflections are separated by ~ 20 ms, and show a clear polarity reversal, which reflects the opposite sign of the acoustic impedance contrast between an ice-water and a water-lake bed transition. The observed shot gather contains a coda following the lake bottom reflection that is absent in the synthetics that do not include a discontinuity at the base of the sediment package (Fig. 5b). When a discontinuity between the sediment and underlying bedrock is included a strong sediment bottom reflection is introduced which more closely matches the observations (Fig. 5c). In the observed data it is difficult to clearly identify a sediment bottom reflection since the complex coda could be caused by reverberations within a thin sediment sequence, or many superposed reflections from individual discontinuities. However, if

the first positive peak following the lake bottom reflection represents the base of the sediment, we can estimate a sediment thickness of 8.5 m assuming a sediment $V_P$ of 1700 m s$^{-1}$ (Table 1).

## 3.2 Lake origin

While our results suggest that L2 is indeed a subglacial lake, its presence is perplexing given its location with a mean annual surface temperature of -22° C and its position beneath a relatively thin column of glacial ice. In contrast to many well studied

subglacial lakes below the Antarctic ice sheet, such as Lake Vostok that lie below ~ 4 km of ice, the basal temperature at our field site is expected to be well below the pressure-dependent melting point of ice. Distinguishing between the different hypotheses of subglacial lake formation has implications for the stability and dynamics of the Greenland ice sheet since they predict different basal thermal and hydrological conditions. Thus, constraining the temperature of L2 is an important goal.

We determine the range of possible basal temperatures using a 1D steady state advection-diffusion heat transfer model solved using the control volume method (see Supporting Information). The modeling assumes an ice density $\rho = 920$ kg m$^{-3}$, a heat capacity $c_P = 2000$ J kg$^{-1}$ K$^{-1}$, and a thermal conductivity of ice of $k = 2.3$ W m$^{-1}$ K$^{-1}$. The basal geothermal heat flux $q$ is varied between $50 - 60$ mW m$^{-2}$, which is consistent with estimates derived from magnetic data (Martos et al., 2018) and thermal isostacy modeling (Artemieva, 2019). Fig. 6 shows results for surface temperatures T$_S$ of $-20°$ C and $-22°$ C and ice-

equivalent accumulation rates $w$ ranging from 0 to 0.3 m yr$^{-1}$ ice equivalent. When vertical advection is ignored (i.e., no ice accumulation), most scenarios predict frozen bed conditions with the exception of the relatively warm surface condition (T$_S$ = $-20°$ C) and high heat flow ($q = 60$ mW m$^{-2}$) scenario (Fig. 6a). When ice accumulation is considered, all scenarios predict frozen bed conditions (Fig. 6b). For an ice- equivalent accumulation rate of 0.3 m yr$^{-1}$, which most closely matches the conditions of the field site, and regional average geothermal flux the basal temperature is expected to be between approximately

240    -12° C and -14° C.

There are several possible explanations for the existence of liquid water underneath the ice, including hypersalinity, recharge by surface meltwater, high geothermal flux, and latent heat from freezing. Here, we review these explanations and assess their specific relevance to lake L2.

(1) Hypersalinity: If the lake is hypersaline the lakewater could remain liquid at low temperatures by depressing the freezing temperature. In order to depress the freezing temperature of water by 12° C to 14° C a NaCl concentration of roughly 160 to 180 ppt would be required, 6x that of seawater (e.g., Fofonoff & Millard Jr, 1983). If the hypersaline condition is restricted to the lake, the surrounding ice would likely be frozen to the bed and would form a closed hydrologic system that could remain

isolated on geologic timescales. In this scenario, the lake could represent a body of ancient marine water that was trapped as glacial ice advanced over the area and potentially further enriched in salt through cryogenic concentration processes (Lyons et al., 2005, 2019). Similar hypersaline lakes with salt concentrations several times higher than sea water are known to exist

below the McMurdo Dry Valleys in Antarctica (Hubbard et al., 2004; Lyons et al., 2005, 2019; Mikucki et al., 2009) and in the Devon Ice Cap, Canada (Rutishauser et al., 2018). Because the current elevation of the lake is more than 500 m above sea level, it is unlikely to be trapped sea water as in the McMurdo Dry Valleys. While an ancient evaporite deposit is possible, as is proposed for the Devon Ice Cap (Rutishauser et al., 2018), the geologic map of Greenland does not indicate likely evaporites in this area (Dawes, 2004).

(2) Surface meltwater: The lake may be part of an open hydrological system that is continually recharged by surface meltwater. If the hydrological system is connected and the rate of recharge matches or exceeds the rate of freezing, a lake could persist despite sub-freezing temperatures in the lower part of the ice. At other locations in Greenland, observations of vertical surface deformation and collapse features have suggested that surface meltwater plays a prominent role in subglacial lake formation and dynamics (Palmer et al., 2015; Willis et al., 2015). This lake, however, is in the high elevation accumulation area of the ice sheet, near the ice divide (Fig. 1b) and there are no obvious sources for significant surface recharge visible on the ground or from satellite imagery. To determine possible pathways for surface recharge from more distant feature, we estimate the local hydraulic head based on surface and bed elevations (Fig. S5) and find no pathways given the present resolution of bed and surface topography. It is possible that a subglacial pathway exists that is smaller than the resolution of BedMachine (Morlighem et al., 2017).

(3) High geothermal flux: Anomalously high basal heat flux may promote melting of the ice sheet from below (e.g., Fahnestock et al. 2001; Rogozhina et al., 2016). If this is the case, the local geothermal heat flux must greatly exceed regional estimates of the geothermal heat flux beneath the northwestern Greenland ice sheet, which are typically in the range of $50 - 60$ mW m$^{-2}$ (Artemieva, 2019; Martos et al., 2018; Rogozhina et al., 2016). Based on the one-dimensional model shown in Fig. 6, a geothermal flux on the order of 100 mW m$^{-2}$ would be necessary to sustain the lake. While high heat flux in this region is unexpected based on the cratonic bedrock geology and lack of recent volcanism, a local region of high heat flux could be promoted by the presence of upper crustal granitoids rich in radiogenic heat producing elements or hydrothermal fluid migration through pre-existing fault systems (e.g., Jordan et al., 2018).

(4) Latent heat from freezing. For the isolated lake of actively freezing brine (as in Hypothesis 1), the hydrologically connected continuous flow (Hypothesis 2), or if the lake is a relic of a larger freshwater body that is slowly freezing, the thermal profile of the ice would show a curvature change at depth due to a latent heat source at the bottom boundary. Given a latent heat of freezing of 334 J g$^{-1}$, freezing a layer 1 m thick to the bottom of the ice over one year is roughly equivalent to increasing the geothermal flux by 10 mW m$^{-2}$.

Sustaining a freezing rate of several m yr$^{-1}$ to generate the latent heat necessary to maintain warm basal ice is less likely than locally elevated geothermal anomaly. We, therefore, narrow the lake origin hypotheses to either anomalously high geothermal

flux or hypersalinity due to local ancient evaporite. Measuring the thermal profile and vertical velocity and strain rates above this lake would provide important information to assess these hypotheses. For a freshwater lake created by high geothermal flux, the basal ice temperature would be near 0° C, vertical velocity would be downward if melting exceeds accumulation. For a lake created by evaporite, the basal ice would be substantially below zero, the vertical velocity would be near zero or upward (due to freezing). A geothermally created lake would show higher vertical strain rates in the lower part of the ice column than an evaporite-created lake.

A freshwater lake and a hypersaline lake have different physical properties and thus may have different signatures that could be detected in geophysical surveys. Radar reflections from an ice/brine boundary undergoing freezing and cryoconcentration of the brine is known to cause scattering and decrease the reflectivity (Badgeley et al., 2017) which we do not see in our data; this provides a second justification to rule out modern active cryoconcentration; in addition, sustained freezing of any ice is likely to create a radar-detectable basal ice unit such as suggested by Bell et al. (2014).

Further, because the seismic velocity and density of water depends on temperature and salinity, we would expect that lakes formed by different mechanisms would have slightly different basal reflection coefficients, although the small variations expected in $c_R$ would not be resolvable with our dataset. On the other hand, because the electrical resistivity of water is strongly dependent on salinity, magnetic sounding could provide useful constraints on lake composition. Additionally, since radar attenuation is strongly sensitive to lake conductivity, radio-echo sounding amplitude data could potentially help constrain salinity if lake bed returns are observed in shallow areas. Stronger constraints could potentially be placed on subglacial properties if a stronger active source were used (e.g., explosives), since high signal to noise ratio data could be recorded at larger distances. This would be particularly useful for measuring the basal reflectivity as a function of incidence angle, which would help verify our interpretation of a subglacial lake. Repeated seismic reflection or GPR surveys calculated along the same transect could provide clues into whether or not lake levels are changing over time (e.g., Church et al., 2020). Finally, direct sampling with drilling would provide the best measurements on subglacial lake properties and could also yield useful biological and paleoenvironmental information.

## 4 Conclusions

We conducted an active source seismic reflection and GPR survey in northwestern Greenland above a site that was previously identified as a possible subglacial lake. We observed a horizontal reflector across the majority survey with a seismic reflection coefficient of -0.43 +/- 0.17, consistent with the presence of a lake below approximately 830 – 845 m of ice. Additionally, we observed a lake bottom reflection near the center of our seismic profile consistent with a lake depth of approximately 15 m. From previous observations of the lateral extent of the lake based on airborne radio-echo sounding (Palmer et al. 2013), we estimate the subglacial lake holds a total of 0.15 Gt of water. Strong coda arriving after the lake-bottom reflection suggests

that the lake is underlain by a sedimentary package but its thickness and material properties are uncertain. To the authors knowledge, this is the first time a ground-based geophysical survey has confirmed the existence of a subglacial lake in Greenland and provided constraints on its depth. Understanding the nature and origins of recently detected subglacial lakes in Greenland is important since wet basal conditions enable glacial ice to flow more easily which can further promote ice loss. Our analysis of the seismic, radar, as well as thermal and hydropotential analysis narrow the lake origins to either locally high geothermal flux or an ancient evaporite deposit. Future work, such as additional geophysical investigations or drilling expeditions, should focus on constraining the temperature and salinity of the lake which will provide clues to its origin.

## Code availability

All seismic processing performed in this study was performed using Obspy (Beyreuther et al., 2010), which is an openly available Python-based software package.

## Data availability

Active source seismic data, including a full description of the dataset, is available through the Digital Repository at the University of Maryland (http://hdl.handle.net/1903/27042). GPR data is available on request from the corresponding author.

## Author contributions

This project was conceptualized by the SIIOS (Seismometer to Investigate Icy Ocean Worlds) team. Analysis of the seismic data was performed by RM and CG with support from NS and KR. EP performed the GPR analysis and thermal modeling. RM was the lead of manuscript writing and figure preparation, with significant inputs from EP, NS, and KR. Authors DD, NW, BA, AM, NH were essential for data collection and curation. JB, VB, and SB were project administrators and also provided critical review and commentary.

## Acknowledgements

Funding for this work was provided by the NASA Planetary Science and Technology Through Analog Research (PSTAR) Grant Number 80NSSC17K0229. The authors thank SIIOS (Seismometer to Investigate Icy Ocean Worlds) team members Chris Carr and Renee Weber for helpful discussions. Additionally, we thank editor Evgeny Podolskiy, two anonymous reviewers, and Jacob Buffo for feedback that helped improve this study. Logistical support for field work in northwestern Greenland was provided by Susan Detweiler.

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

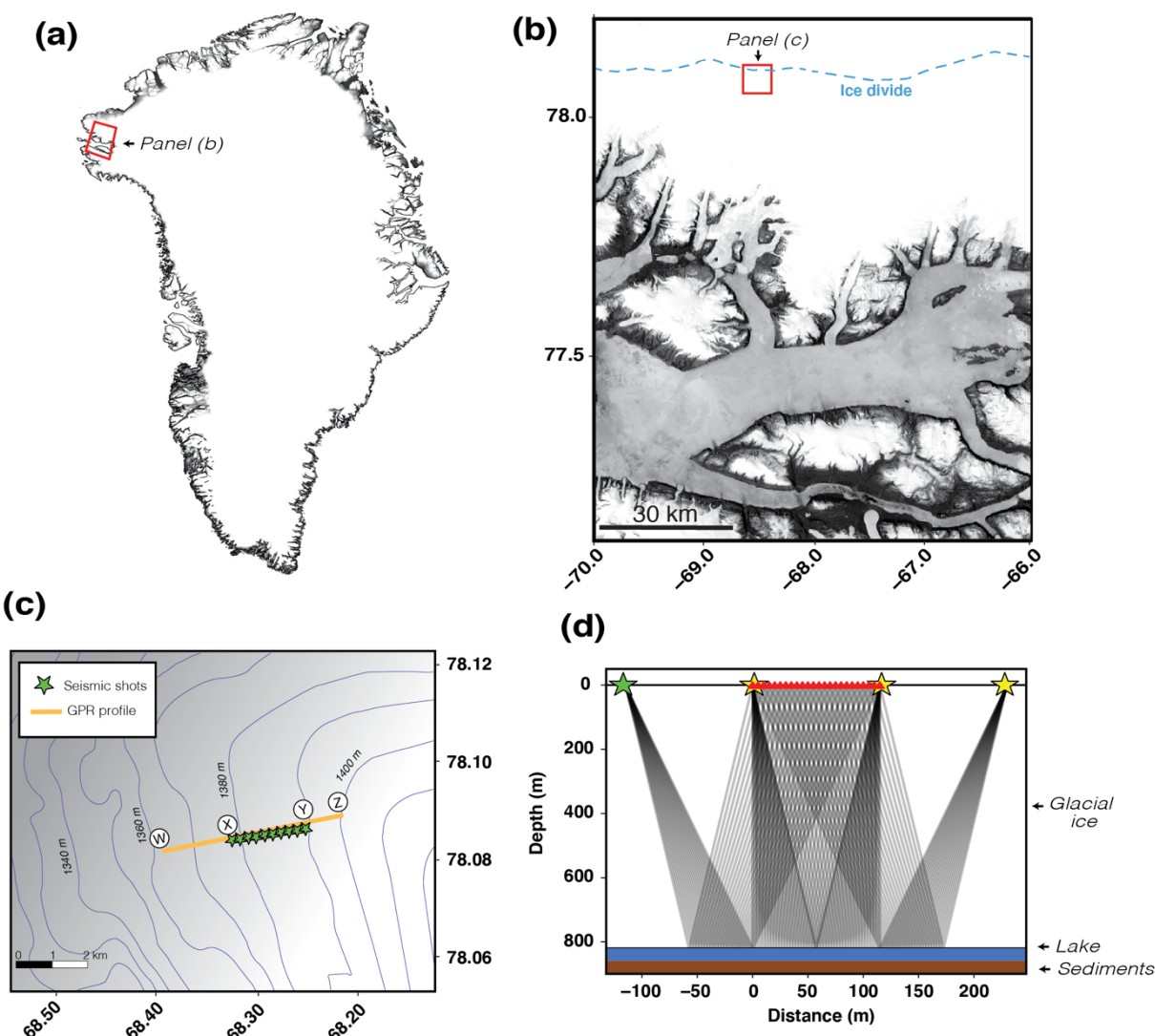

Figure 1: (a) Map of Greenland showing our field location in the northwest. (c) Composite satellite image from Landsat 8 taken between 2018-5-20 and 2018-5-27. (c) Close up map of field region. The green stars show the active source shot and the orange line shows the track of the GPR survey. Only the first of 4 shot locations for each geophone line is plotted. (d) Geometry of the active source experiment for a single geophone line. The black lines indicate the raypaths of R1 between all source locations (stars) and

geophones (red triangles).

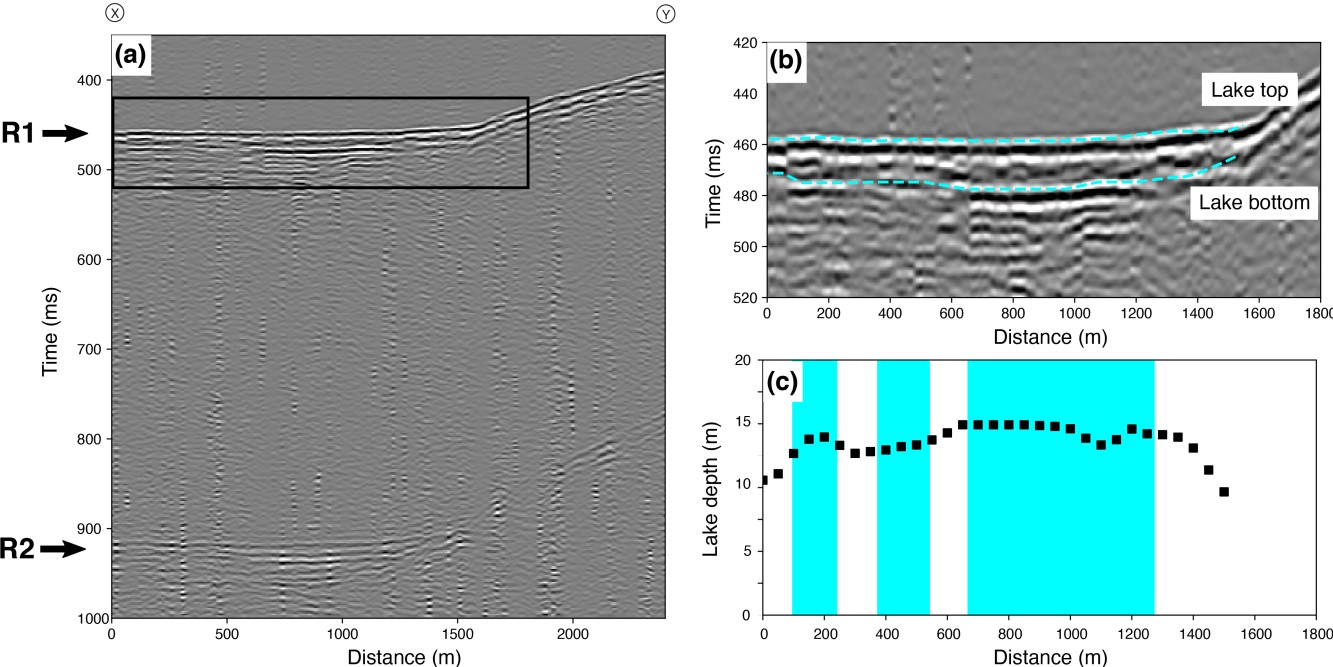

**Figure 2. (a) shows the seismic reflection profile of the entire traverse. Reflections labeled R1 and R2 correspond to the primary reflection from the lake top and its multiple. A transect distance of 0 m corresponds to the southwestern end of the line. (b) shows a close up of the R1 reflection window (black rectangle in (a)), showing reflections from both the lake top and bottom. Travel time picks of the lake top and lake bottom reflections are drawn with the dashed blue line. The depth of the lake inferred from the picked reflections assuming a lake $V_P$ of 1498 m s$^{-1}$ is shown in (c). Blue shaded regions indicate where the lake bottom reflection is most clearly identified.**

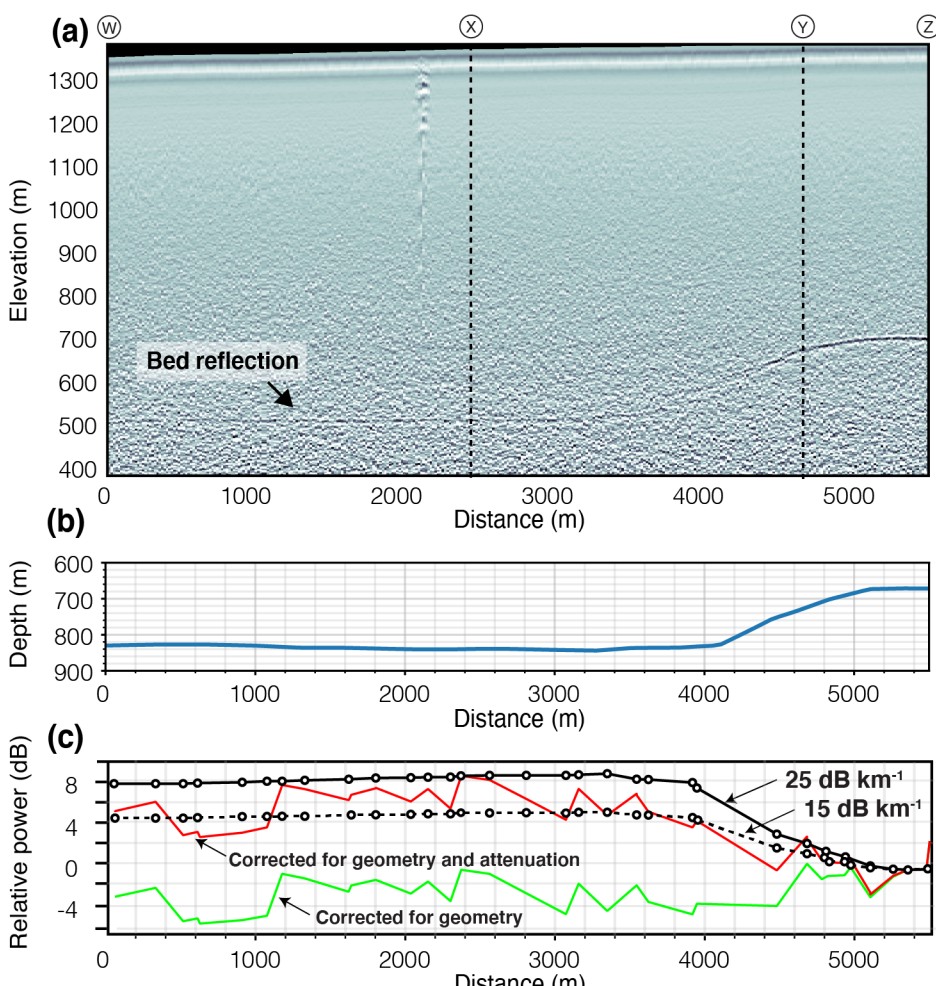

**Figure 3. GPR profile. (a) shows the 5 MHz radar data, unmigrated. The primary bed reflection is marked with an arrow. Vertical dashed lines mark the approximate endpoints of the seismic survey. The depth from the surface to the base of the ice is shown in (b). In panel (c), the relative power of the basal reflections is shown after being corrected for geometric spreading (green line) and both geometric spreading and depth-average attenuation of -15 dB km⁻¹ (red line). The black solid and dashed lines show the magnitude of attenuation corrections assuming an englacial attenuation of -25 dB km⁻¹ and -15 dB km⁻¹, respectively.**

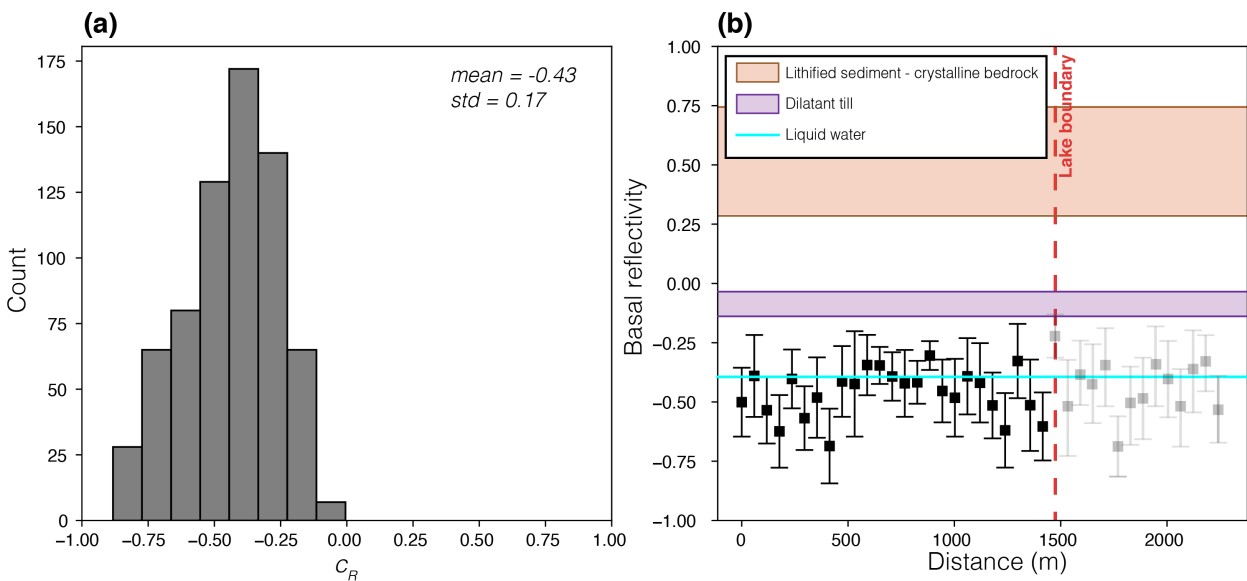

**Figure 4. (a) Distribution of reflection coefficients $c_R$ calculated for all shots in the survey. (b) Basal reflectivity as a function of distance along the transect. The black scatter points with error bars show the mean and standard deviation of $c_R$ in a single shot gather, calculated assuming an absorption factor a = 0.23. The shaded regions show the range of expected basal reflectivity values for bedrock or dilatant till and the cyan line shows the basal reflectivity expected for liquid water. The approximate boundary of the subglacial lake is marked by the red dashed line. Values beyond the margin of the lake are shown with light shading because they cannot be confidently interpreted due to the low signal strength of the R2 reflection.**

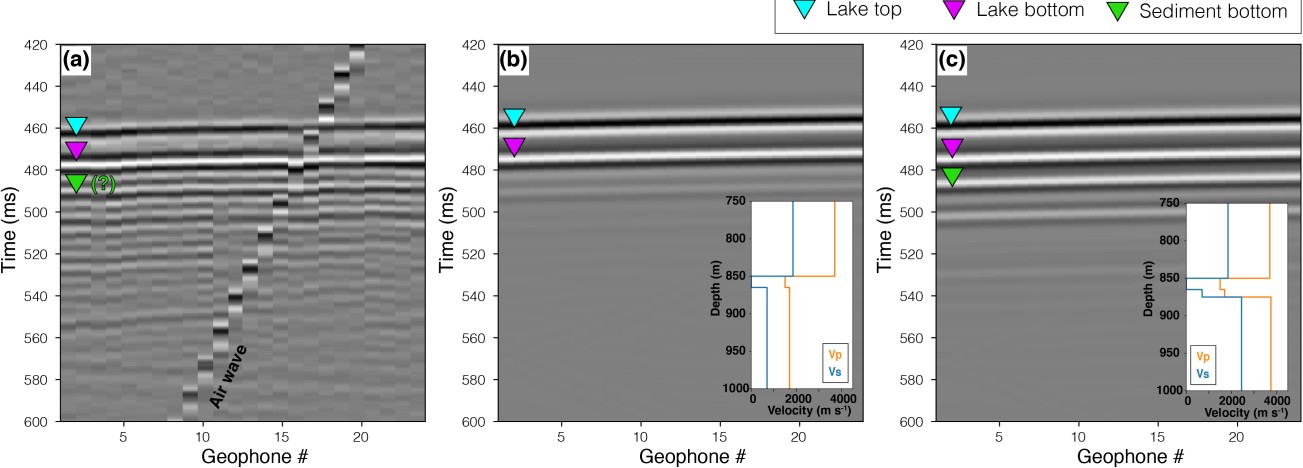

**Figure 5. Observed (a) and synthetic (b and c) seismic data for shot gather 12 bandpass filtered between 50 – 200 Hz. The offset from the source to geophone 1 is 230 m. The colored triangles indicate reflections from the lake top (blue), lake bottom (purple), and sediment bottom (green). The insets in panels (b) and (c) show the $V_P$ and $V_S$ models that were used to compute the synthetics. Both models include a 12 m thick lake below 850 m of ice. The model used in (c) includes an additional discontinuity 10 m below the lake, which represents the boundary between the lake bottom sediments and underlying bedrock.**

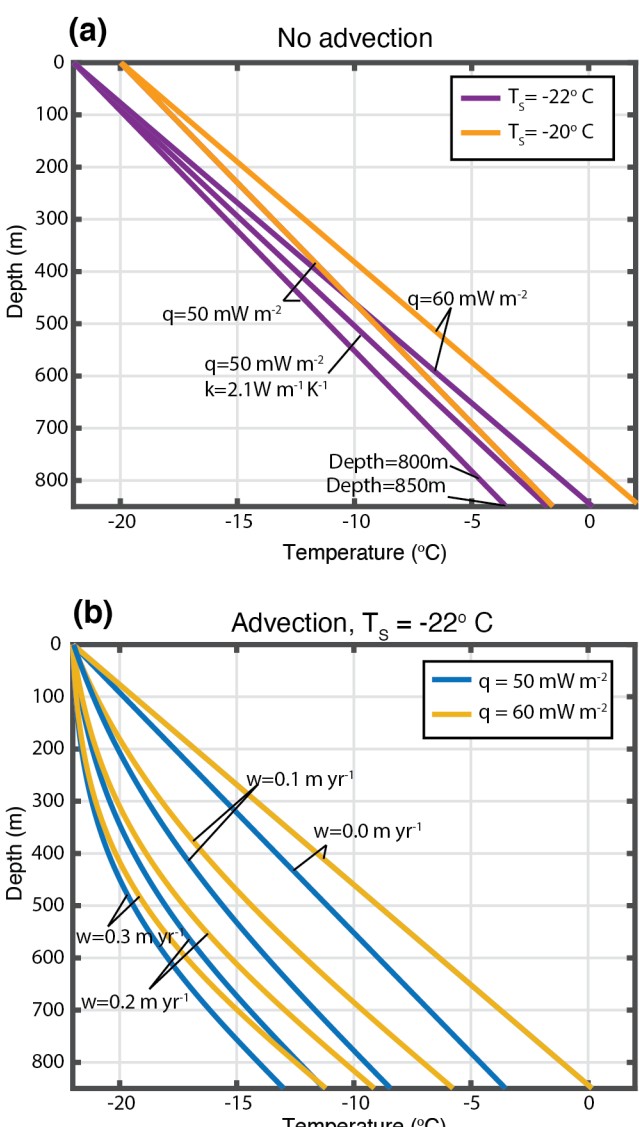

**Figure 6. Modeled ice sheet thermal structure. Panel (a) shows thermal profiles neglecting advection for surface temperatures $T_S$ = -22° C and $T_S$ = -20° C. Panel (b) shows thermal profiles including advection for a fixed surface temperature of $T_S$ = -22° C. The basal heat flux q is varied between 50 – 60 mW m$^{-2}$ and the accumulation rate w is varied between 0 m yr$^{-1}$ and 0.3 m yr$^{-1}$.**

| Material | $V_P$ (m s$^{-1}$) | $V_S$ (m s$^{-1}$) | Density (kg m$^{-3}$) |
|---|---|---|---|
| Glacial ice | 3810[a] | 1860[a] | 920[a] |
| Water | 1498[a] | 0 | 1000 |
| Dilatant sediment | 1600 –1800[b] | 100 – 500[b] | 1600 – 1800[b] |
| Lithified sediment | 3000[b] – 3750[a] | 1200[b] – 2450[a] | 2200[b] – 2450[a] |
| Bedrock | 5200[a] – 6200[b] | 2700[a] – 3400[b] | 2700[a] – 2800[b] |

**Table 1. Description of material properties used in reflection coefficient modeling. Values are compiled from Peters et al. (2008)[a], and Christianson et al. (2014)[b].**