# Peer review of "Geophysical constraints on the properties of a subglacial lake in northwest Greenland"

_The Cryosphere, 2020_

## Referee Comment (RC1) · Anonymous Referee #1 · 14 Nov 2020

This manuscript presents a targeted geophysical study of a Greenland subglacial lake which combines seismic and GPR data to constrain the physical properties of a subglacial lake. The seismic portion of the analysis is largely persuasive and provides a depth measurement for a lake that had previously been detected by airborne radar sounding. However, the GPR portion of the analysis is underdeveloped. Specifically:

- The reflectivity of the subglacial lake in the GPR data was not presented or analyzed at a sufficient level of depth. This should be compared and combined with the seismic data to provide a more complete and quantitative picture.

- The GPR reflectivity signal is also not sufficiently compared to the airborne radar sounding data in the Palmer paper that motivated the study or to more recent radar studies of Greenland subglacial hydrology and thermal state like: Chu, W., et al. "Complex basal thermal transition near the onset of Petermann Glacier, Greenland." Journal of Geophysical Research: Earth Surface 123.5 (2018) or Jordan, T., et al. "A constraint upon the basal water distribution and thermal state of the Greenland Ice Sheet from radar bed echoes." Cryosphere (2018).

- One of the most exciting opportunities from combining seismic and GPR data over lakes is the ability to use the sensitivity of radar attenuation to water conductivity, which the authors mention in the context of EM surveys, to constrain lake salinity/conductivity. The paper would be stronger if this was included.

---

## Short Comment (SC1) · 25 Nov 2020

Hello,

First and foremost congratulations and excellent work on a paper which utilizes an array of geophysical methods to probe the existence and properties of a subglacial hydrological feature that has broad implications for a number of scientific fields, including glaciology, climate science, and planetary science.

I do however have a number of comments and concerns with the current manuscript:

1) I believe in the current manuscript the geothermal heat flux labels of Figure 6b are mislabeled and need to be switched.

[Figure]

2) I do not feel the 1D thermal model of the ice sheet is described in enough detail so as to reproduce or validate the presented results. There is a broad reference to Patankar (1980) but this text focuses on general numerical methods rather than the setup for the specific ice sheet problem discussed here. What is the advection term utilized here? Is it the deposition rate? Accumulation rates are given in 'ice equivalent' form, but are these deposited at the already compacted ice density of 920 kg/mˆ3 or at a lower density and then compacted? I think expanding on the description of the model would help to clarify the utility of the results.

3) At no point are the reflectivity results gathered over the presumed lake (either GPR or seismic) quantitatively compared to the surrounding bedrock reflectivity values. This seems like a missed opportunity to me. The difference in expected reflectivity between bedrock and an ice-water phase transition is discussed, and hypothetical reflection coefficients are plotted in Figure 4, however it is not demonstrated that this is observed in the current study site. I find results comparing such contrasts in reflectivity crucial to the validity of these types of studies - for example Rutishauser et al (2018) "Discovery of a hypersaline subglacial lake complex beneath Devon Ice Cap, Canadian Arctic" present relative power measurements that show striking contrast between regions with lakes and the surrounding bedrock. I feel a comparable approach could be taken in this manuscript to substantially bolster the evidence for the existence of a lake. I do not feel qualitative inspection of the radargram in Figure 2 is enough evidence to conclude that a lake is present. Why are reflection coefficients for regions not directly over the lake excluded from Figure 4 (when this could validate the claims made in the manuscript)? Without an explicit example of contrasting properties between the purported lake and surrounding terrain I do not feel that the conclusion of a substantial (10-15 m thick) lake existing beneath the ice is a valid one.

Best,

Jacob Buffo, PhD Dartmouth College Thayer School of Engineering

---

## Referee Comment (RC2) · Anonymous Referee #2 · 14 Dec 2020

1. General comments:

This paper reports seismic and ground-based radar measurements on a subglacial lake in northwestern Greenland. The target of the study is the subglacial lake firstly discovered in Greenland in 2013 based on airborne radar measurements. Some other radar surveys were performed at other lakes beneath the Greenland ice sheet, but this is the first ground-based lake observation in Greenland. The seismic signals enabled the authors to quantify the water depth of the lake, as well as to estimate the material underneath the lake. One dimensional thermal analysis suggested that the lake is filled with hypersaline water under a condition of well below water freezing temperature. In contrast to increasing number of studies on subglacial lakes in Antarctica, much less is reported and known about those in Greenland. The seismic data presented in this

paper are valuable, because they provide information below ice-water surface, namely water depth and lake-bed constitution. Similar studies have been performed at some lakes in Antarctica, but this is the first case in Greenland. Numerical analysis of ice temperature is simple, but enough to provide insights into lake water composition and origin of the lake formation. Because of these reasons, I think the reported data are valuable and of great interests of the journal readers. The paper is clearly written. However, it is too concise in some parts and essential information is missing. In general, my impression is that details of method, data and analysis are not sufficiently presented as expected in a paper published in this journal. I am also concerned about the structure of the sections. I list below my concerns, which are followed by more specific comments and corrections. I hope they are considered to improve the paper.

2. Major concerns

(1) Presentation of methodology Some essential information is missing about the measurements and analysis used in this study. For example, radar device is described only by "a 10 MHz monopulse radar system". Information of the manufacturer, type of antenna, receiver-transmitter distance, the way of data acquisition and dragging the device (sledge?) should be described. Another example is ice temperature analysis. Only available information for this computation is "1D steady state advection-diffusion heat transfer model solved using the control volume method". How do you compute vertical strain rate? What is spatial resolution? Any influence of neglected firn layer and horizontal advection? Please describe all these details in the Method section.

(2) Presentation of data Results of the seismic and radar survey are presented in a limited way (Figs 2, 3 and 5). They are given only by plotting amplitude or power in a grey scale on a time-space domain. I wonder how the authors determine reflections at ice-water and water-bottom reflections. Fig. 2B and Fig 3A show important boundaries, but it gives me an impression that they were drawn only by visual inspection. Further, the authors discuss the phase of the seismic signals to identify the material under the ice. Nevertheless, there is no plot clearly showing such an important observation. I

think more details, particularly plots of amplitude/power against time, are necessary to convince the readers of the interpretations and discussions.

(3) Comparison with previous studies Seismic survey on a glacial lake is new in Greenland, but available for lakes beneath the Antarctic ice sheet. Interpretation of the seismic signals should be carried out based on the knowledge obtained in Antarctica. Such studies in Antarctica include those reported in Whillans Ice Stream and Lake Ellsworth. Important previous work exists also in Devon Ice Cap in the Canadian Antarctica. Considering the proximity of the sites and possible similarity in water property, closer comparison of the thermal conditions, geographical and geological settings should be performed. Please also introduce these previous studies more in detail in the Introduction section. I would like to read what are known about water depth, lake-bed constitution, water properties in subglacial lakes in Antarctica and other regions.

(4) Construction of the sections The paper suffers from mixing of method, results and discussion in the text, particularly in the Methods section. The Methods section begins with study site, and a little of methodology of seismic and GPR measurements (2.1 Field experiment). Then, it explains a bit more about the seismic measurement and directly goes into data and interpretation (2.2 Seismic and GPR imaging). Next subsection explains the analysis of the reflection power, which is followed by interpretation of the data (2.3 Basal reflectivity). This is not usual as a journal article and not convenient for readers. Please consider reconstruction of the text. The best for readers is to explain all the methodology in the Method section, which is followed by presentation of data in detail but without interpretation in the Results section, and finally interpretation and discussion in the Discussion section. I also find the last paragraph of the Introduction section includes too much results and conclusion. I would expect this kind of summary of the study in Abstract, which is currently rather weak.

3. Specific comments:

Line 15-20: This abstract can be improved by incorporating the essential results of the

measurements and conclusion described in the last paragraph of Introduction (Line 68-77).

Line 32: "Bentley et al., 2011" » Missing in the reference list (or the publication year is wrong).

Line 36-37: "airborne radio-echo sounding" » "airborne" is not a necessary condition. Snow vehicle or snow mobile are also used for surveying lakes.

Line 51: "approximately 40% of ..." » This is not consistent with 124 out of 400 as described in Line 40.

Line 68-77: I think this is too much for Introduction. Please consider moving the essential results in Abstract.

Line 81: Please provide coordinates and elevation of the lake.

Line 82: Can you indicate the 980 km2 drainage basin on Figure 1B?

Line 86: "24 40 Hz" » Hyphen is missing.

Line 96: "longitudinal seismic reflection image" » Here and other places, the authors use "longitudinal" and "across", which are not clear to explain settings. Here, for example, "seismic reflection image along the survey route" is better if I understand it correctly.

Line 107-108: "An additional reflection with opposite polarity of R1" is not clearly shown by Fig. 2B. Also not clear why you think "which is consistent with a lake bottom reflection".

Line 110: What do you mean by "across the seismic section"?

Line 111-112: Uncertainty due to wave velocity is evaluated, but I wonder if there is additional uncertainty due to signal peak determination. How do you define the reflection boundaries in Figure 2B?

Line 116: "across the majority of the transect" » "across" is confusing.

Line 117-118: "lake is slightly deeper" » Do you mean "ice is slightly thicker"?

Line 118: Please define "transect distance".

Line 126: "A_R1 and A_R2" » The variable "A" should be in italic?

Line 160: "IMBIE Team Report" » The author name is inconsistent with the reference list.

Line 160: Can you provide an estimate of "net storage capacity of all of Greenland's subglacial lakes"?

Line 180: How do you know the surface temperature in the region?

Line 181-182: "the basal temperature ... be well below the pressure dependent melting point" » Why do you think so?

Line 185: "1D steady state advection-diffusion heat transfer model" » Please describe more details with equations to be solved.

Line 190: "When advection is ignored" » I understand that you ignore vertical ice motion. It is confusing because you also neglect horizontal advection of ice. Ice flow is small near the divide and downglacier advection of cold ice does not influence the conclusion about basal temperature below melting point, but mentioning the horizontal ice flow helps the readers.

Line 199: Do you have estimate of the salinity from the computed basal temperature? Can you discuss your results with the study at Devon Ice Cap?

Line 199-200: "ice surrounding the lake would be frozen" » Do you think the hypersaline condition is limited with in the lake area? Such condition may extend to the surrounding area and cause basal melting outside of the lake.

Line 210-219: I agree that continuous supply of surface meltwater to the bed is not likely

because meltwater production is limited in this elevation range. Near the study site, a Japanese research group has been running an automatic weather station (e.g. Aoki et la., 2014), performed in-situ snow observations and ice core studies (e.g. Niwano et al., 2015; Kurosaki et al., 2020). I suggest the author to discuss water availability in the region based on the climatic conditions and the previous studies.

- Aoki, T. et al. (2014). Field activities of the "Snow Impurity and Glacial Microbe effects on abrupt warming in the Arctic" (SIGMA) Project in Greenland in 2011-2013. Bulletin of Glaciological Research. 32. 3-20. 10.5331/bgr.32.3. - Niwano, M. et al. (2015). Numerical simulation of extreme snowmelt observed at the SIGMA-A site, northwest Greenland, during summer 2012. The Cryosphere. 9. 2015. 10.5194/tc-9-971-2015. - Kurosaki, Y. et al. (2020). Reconstruction of Sea Ice Concentration in Northern Baffin Bay Using Deuterium Excess in a Coastal Ice Core From the Northwestern Greenland Ice Sheet. Journal of Geophysical Research: Atmospheres. 125. 10.1029/2019JD031668.

Line 235: "cryoconcentration" » Is this a right word to explain lake formation due to "latent heat from freezing".

Line 235-239: It is odd to read this conclusion within the same paragraph explaining "Latent hear from freezing". Please consider to change the paragraph, or merge these sentences with the next paragraph.

Line 266: "Peters et al., 2013" » Missing in the reference list (or the publication year is wrong).

Line 272: "hydropotential modeling" » "hydropotential analysis"?

Figure 1c: Please label the ends of the GPR and seismic survey profiles (e.g. "X" and "Y") so that you can use the labels on Figures 2 and 3.

Figure 6B: There is something wrong with the line colors. I would expect warmer temperature for the higher geothermal heat flux.

Figure S2B: The vertical axis label "Ice Sheet Velocity" is odd. It's seismic wave velocity, right?

Figure S4: Please enlarge the study site and consider drawing contour lines. Otherwise, the color scale map does not tell a lot about the hydraulic potential distributions around the lake.

---

## Author Comment (AC1) · 6 Feb 2021

We agree that the GPR analysis needed improvement. We have added more details to Section 2.1 to clarify the data collection and processing methods. The new GPR Methods section reads as follows:

*"The GPR data was collected across a  5.5 km transect roughly parallel to the seismic survey (Fig. 1C), using an acquisition system specially adapted to be towed by a motor sled traveling at approximately 10 km/hr (e.g., Welch  Jacobel, 2003).  The system used a Kentech pulse transmitter that produces +/- 2000 V pulses with a variable pulse repetition frequency of between 1 and 5 kHz. The antennae are resistively loaded wire dipoles with nominal frequency of 5MHz, and the receiver uses an 8-bit NI USB-5132*

*digitizer and a computer. Between 16 and 64 radar shots were stacked were stacked and filtered 2 to 8MHz to produce each final trace on the radargram. A GPR reflection image was created by converting the radar data to depth using a radar velocity of 172 $m/\mu s$."*

Additionally, we have updated the radar image in Figure 3 which now shows a sharper picture of the ice bottom reflection (see attached). Finally, we included some analysis of the ice bottom reflectivity. We find that the reflectivity above the lake is approximately 10 dB stronger than the surrounding region, which is broadly consistent with the results of Palmer et al (2013), who found a variation of between 10 - 20 dB. However, we disagree that a detailed discussion of the differences with Palmer et al. (2013) would be useful. Comparing radar reflectivities obtained from multiple surveys conducted at different times (and here different collection methods) is challenging and only rarely done (e.g., Schroeder, D., Hilger, A., Paden, J., Young, D., Corr, H. (2018). Ocean access beneath the southwest tributary of Pine Island Glacier, West Antarctica. Annals of Glaciology, 59(76pt1), 10-15. doi:10.1017/aog.2017.45).

Using the radar reflections to constrain salinity would be an exciting possibility. However, in our GPR results, there are no clear returns from signals that have traversed the lake (i.e., lake bottom reflections). This is likely because the water layer is too highly attenuating. The highest likelihood of detecting lake bottom reflections may be near the edge of the lake where the water layer is thin, yet we can not confidently interpret any signals beyond the primary ice bottom reflection near the lake boundary. We have added some discussion about this to the manuscript.
* * *
**A**

bed reflection

**B**

Fig. 1.

---

## Author Comment (AC2) · 6 Feb 2021

We thank the reviewer for their thorough review, and will respond to their comments point by point below. Comments by the reviewer are bolded.

**(1) Presentation of methodology Some essential information is missing about the measurements and analysis used in this study. For example, radar device is described only by "a 10 MHz monopulse radar system". Information of the manufacturer, type of antenna, receiver-transmitter distance, the way of data acquisition and dragging the device (sledge?) should be described. Another example is ice temperature analysis. Only available information for this computation is "1D steady state advection-diffusion heat transfer model solved using the control**

**volume method". How do you compute vertical strain rate? What is spatial resolution? Any influence of neglected firn layer and horizontal advection? Please describe all these details in the Method section.**

We have added a more detailed description of the GPR data acquisition and processing to the Methods section (see also our response to Reviewer 1). Additionally, we have added a new section to the Supporting Information that provides a detailed description of the thermal modeling, along with its assumptions. See the attached document for more details.

**(2) Presentation of data Results of the seismic and radar survey are presented in a limited way (Figs 2, 3 and 5). They are given only by plotting amplitude or power in a grey scale on a time-space domain. I wonder how the authors determine reflections at ice-water and water-bottom reflections. Fig. 2B and Fig 3A show important boundaries, but it gives me an impression that they were drawn only by visual inspection. Further, the authors discuss the phase of the seismic signals to identify the material under the ice. Nevertheless, there is no plot clearly showing such an important observation. I think more details, particularly plots of amplitude/power against time, are necessary to convince the readers of the interpretations and discussions.**

The seismic reflections analyzed here (i.e., from either the ice or lake bottom) are clearly distinguishable from other phases based on their moveout; reflected energy arrives at all geophones in the line nearly simultaneously due to the near vertical incidence. In other words, there is no ambiguity of whether or not the phases we are interpreting are subsurface reflections. The reviewer states that it appears that the ice bottom and lake bottom reflections appear to have been drawn based only on visual inspection, yet this is common practice and well accepted. Additionally, we are not certain what is meant by "results of the seismic and radar survey are presented in a limited way", since the cross sections shown in Figures 2A and 3 show all of the relevant data collected from the seismic and GPR surveys, respectively. The presentation

style (i.e., using grey scale colormaps) is a very common way of displaying reflection results, and to us, seems to be a matter of personal preference. However, we do agree that the phase of the seismic reflections (particularly the opposite polarity of R1 with respect to the seismic source), was not clearly demonstrated. Thus, we have added a new section to the supplement that outlines the polarity analysis, and we include a new figure that more clearly demonstrates the opposite polarity of the source and reflection. The figure, which will be our new Fig. S4, is attached below. The waveforms shown in black in panels B and C are the first arrival and R1 reflection, respectively. In panel C, we have rescaled and reversed the polarity of the first arrival, and aligned it with R1 (shown in grey).

**(3) Comparison with previous studies Seismic survey on a glacial lake is new in Greenland, but available for lakes beneath the Antarctic ice sheet. Interpretation of the seismic signals should be carried out based on the knowledge obtained in Antarctica. Such studies in Antarctica include those reported in Whillans Ice Stream and Lake Ellsworth. Important previous work exists also in Devon Ice Cap in the Canadian Antarctica. Considering the proximity of the sites and possible similarity in water property, closer comparison of the thermal conditions, geographical and geological settings should be performed. Please also introduce these previous studies more in detail in the Introduction section. I would like to read what are known about water depth, lake–bed constitution, water properties in subglacial lakes in Antarctica and other regions.**

We have included a more thorough summary of previous active source seismic surveys conducted in Antarctica, which provides more context for the present study. However, we point out that many of these regions have fundamentally different geological histories and properties, and there is no strong reason to expect any similarity in subglacial water properties. Therefore, direct comparisons are challenging.

**(4) Construction of the sections The paper suffers from mixing of method, results and discussion in the text, particularly in the Methods section. The Meth-**

**ods section begins with study site, and a little of methodology of seismic and GPR measurements (2.1 Field experiment). Then, it explains a bit more about the seismic measurement and directly goes into data and interpretation (2.2 Seismic and GPR imaging). Next subsection explains the analysis of the reflection power, which is followed by interpretation of the data (2.3 Basal reflectivity). This is not usual as a journal article and not convenient for readers. Please consider reconstruction of the text. The best for readers is to explain all the methodology in the Method section, which is followed by presentation of data in detail but without interpretation in the Results section, and finally interpretation and discussion in the Discussion section. I also find the last paragraph of the Introduction section includes too much results and conclusion. I would expect this kind of summary of the study in Abstract, which is currently rather weak.**

We agree that a restructuring of the paper was warranted. In the new version the results are clearly separated from the methods in a new section titled Results. Additionally, we have removed the last paragraph of the introduction and moved the main points to the abstract.

**Line 15–20: This abstract can be improved by incorporating the essential results of the measurements and conclusion described in the last paragraph of Introduction (Line 68–77).**

See response above.

**Line 32: "Bentley et al., 2011" > Missing in the reference list (or the publication year is wrong).**

The publication year was wrong in the references and we have corrected it.

**Line 36–37: "airborne radio-echo sounding"> "airborne" is not a necessary condition. Snow vehicle or snow mobile are also used for surveying lakes.**

We have deleted the word "airborne".

**Line 51: "approximately 40% of ..." > This is not consistent with 124 out of 400 as described in Line 40.**

The discrepancy likely comes from different author determinations of "active" subglacial lakes. We have removed the inconsistency, and prefer to reference Smith et al. (2009), who found evidence of 124 active subglacial lakes in Antarctica.

**Line 81: Please provide coordinates and elevation of the lake.**

We have added a table with the latitudes, longitudes, and elevations of our seismic shot locations to the Supporting Information (new Table S1).

**Line 82: Can you indicate the 980 km2 drainage basin on Figure 1B?**

We refer the reader to Figure 1 of Palmer et al. (2013), who provide a plot of regional bed topography.

**Line 86: "24 40 Hz" > Hyphen is missing.**

Fixed.

**Line 96: "longitudinal seismic reflection image" > Here and other places, the authors use "longitudinal" and "across", which are not clear to explain settings. Here, for example, "seismic reflection image along the survey route" is better if I understand it correctly.**

We agree that "longitudinal" was unnecessary, and have removed it.

**Line 107-108: "An additional reflection with opposite polarity of R1" is not clearly shown by Fig. 2B. Also not clear why you think "which is consistent with a lake bottom reflection".**

The opposite polarities of R1 (i.e., the "lake top" reflection) and the secondary reflection we identify as "the lake bottom" reflection is more clearly demonstrated in Figure 5A, which shows a close up view of the lake reflection sequence on a single shot gather

collected above the lake (shot gather number 12). It is expected that lake top and lake bottom reflections will have opposite polarities because of the opposite reflection coefficients between a layer of ice over water (negative reflection coefficient), and a layer of water over sediment (positive reflection coefficient). This is indeed supported by our modeling results shown in Figure 5B and 5C.

**Line 110: What do you mean by "across the seismic section"?**

We have replaced "across the seismic section" with "as a function of distance along the transect".

**Line 111–112: Uncertainty due to wave velocity is evaluated, but I wonder if there is additional uncertainty due to signal peak determination. How do you define the reflection boundaries in Figure 2B?**

This is a good point. There is likely to be variation between different analysts in terms of how they preprocess their data and how they determine phase arrivals and amplitudes, which would introduce some uncertainty. However, this uncertainty is difficult to quantify, and is always prevalent in any such seismic analysis. In this study, we do not believe that the travel time uncertainty would have a large impact on our lake depth results because a difference of several ms in travel time picks would translate to only small changes in the inferred lake depth (less than 5 m or so). When determining the lake thickness, we prefer to pick the first breaks (of R1 and the subsequent 'lake bottom' reflection) in the processed seismic image.

**Line 116: "across the majority of the transect" > "across" is confusing.**

Replaced "across" with "along".

**Line 117-118: "lake is slightly deeper" > Do you mean "ice is slightly thicker"?**

Yes. We have added clarification.

**Line 118: Please define "transect distance".**

The transect distance is the distance along the profiles shown in Fig 1C. We have added labels W, X, ,Y, and Z to the map in Fig 1C and to our cross sections, which should make this clear.

**Line 126: "$A_{R1}$ and $A_{R2}$" > The variable "A" should be in italic?**

We have replaced all instances of $A_{R1}$ and $A_{R2}$ to be formatted consistently with how they are presented in Equation 1.

**Line 160: "IMBIE Team Report" > The author name is inconsistent with the reference list.**

Fixed.

**Line 160: Can you provide an estimate of "net storage capacity of all of Greenland's subglacial lakes"?**

I think that the uncertainties on such an estimate would be too large to make a useful addition. The first–order assumption that would need to be made is that all of the high reflectivity regions identified in airborne radar surveys (e.g., Bowling et al. 2020) represent subglacial lakes with similar depth than the one we identify in this study, which does not seem justifiable.

**Line 180: How do you know the surface temperature in the region?**

The surface temperature is determined from RACMO2 modeling. This has been made explicit in the text (see below).

*"Using RACMO2 1-km resolution modeling of Greenland's near surface climate and surface mass balance (Noel et al., 2018), we estimate the mean annual air temperature to be -22 C. "*

**Line 181–182: "the basal temperature ... be well below the pressure dependent melting point" > Why do you think so?**

[Figure]

Our estimates of basal temperature are determined through the thermal modeling results summarized in Figure 6.

**Line 185: "1D steady state advection–diffusion heat transfer model" > Please describe more details with equations to be solved.**

We have added a new section to the Supporting Information that fully describes the thermal modeling, including the equations that are solved and the assumptions that we make (also see attached document below).

**Line 190: "When advection is ignored" > I understand that you ignore vertical ice motion. It is confusing because you also neglect horizontal advection of ice. Ice flow is small near the divide and down glacier advection of cold ice does not influence the conclusion about basal temperature below melting point, but mentioning the horizontal ice flow helps the readers.**

By "when advection is ignored", we are referring to cases in which ice does not accumulate. When accumulation is considered (i.e., when we consider 'advection'), the thermal profiles are altered because the near surface isotherms are moved to deeper depths. We clarified this in the manuscript.

**Line 199: Do you have estimate of the salinity from the computed basal temperature? Can you discuss your results with the study at Devon Ice Cap?**

Based on our thermal modeling that suggests a basal temperature of between -12 C and -14 C, we estimate that the salinity required to keep the lake liquid would need to be between 160 and 180 ppt. This is comparable to the results of Rutishauser, who suggested a salinity of 140 – 160 ppt for the Devon Ice Cap region. We have added this discussion to the manuscript.

**Line 199-200: "ice surrounding the lake would be frozen" > Do you think the hypersaline condition is limited with in the lake area? Such condition may extend to the surrounding area and cause basal melting outside of the lake.**

[Figure]

This is a good point and we agree that the hypersaline condition may not be limited to the lake. Discussion has been added to Section 3.1 of the manuscript.

**Line 210-219: I agree that continuous supply of surface meltwater to the bed is not likely because meltwater production is limited in this elevation range. Near the study site, a Japanese research group has been running an automatic weather station (e.g. Aoki et la., 2014), performed in-situ snow observations and ice core studies (e.g. Niwano et al., 2015; Kurosaki et al., 2020). I suggest the author to discuss water availability in the region based on the climatic conditions and the previous studies. - Aoki, T. et al. (2014). Field activities of the "Snow Impurity and Glacial Microbe effects on abrupt warming in the Arctic" (SIGMA) Project in Greenland in 2011-2013. Bulletin of Glaciological Research. 32. 3-20. 10.5331/bgr.32.3. - Niwano, M. et al. (2015). Numerical simulation of extreme snowmelt observed at the SIGMA-A site, northwest Greenland, during summer 2012. The Cryosphere. 9. 2015. 10.5194/tc9-971-2015. - Kurosaki, Y. et al. (2020). Reconstruction of Sea Ice Concentration in Northern Baffin Bay Using Deuterium Excess in a Coastal Ice Core From the Northwestern Greenland Ice Sheet. Journal of Geophysical Research: Atmospheres. 125. 10.1029/2019JD031668.**

While it is important to consider water availability, there are no obvious pathways for surface meltwater to recharge the subglacial lake (e.g., moulins / crevasses), even if it were available.

**Line 235: "cryoconcentration" > Is this a right word to explain lake formation due to "latent heat from freezing".**

We agree that this was confusing. We have clarified our meaning. The new text is below.

*"For the isolated lake of actively freezing brine (as in Hypothesis 1), the hydrologically connected continuous flow (Hypothesis 2), or if the lake is a relic of a larger freshwater body that is slowly freezing, the thermal profile of the ice would show a curvature*

*change at depth due to a latent heat source at the bottom boundary. Given a latent heat of freezing of 334 J/g, freezing a layer 1 m thick to the bottom of the ice over one year is roughly equivalent to increasing the geothermal flux by 10 mW/m2."*

**Line 235-239: It is odd to read this conclusion within the same paragraph explaining "Latent heat from freezing". Please consider to change the paragraph, or merge these sentences with the next paragraph.**

Done.

**Line 266: "Peters et al., 2013" > Missing in the reference list (or the publication year is wrong).**

Thank you for catching this. The correct citation is "Palmer et al. 2013".

**Line 272: "hydropotential modeling" > "hydropotential analysis"?**

Done.

**Figure 1c: Please label the ends of the GPR and seismic survey profiles (e.g. "X" and "Y") so that you can use the labels on Figures 2 and 3.**

We have added labels W,X,Y and Z to the map in Figure 1c and the corresponding cross sections.

**Figure 6B: There is something wrong with the line colors. I would expect warmer temperature for the higher geothermal heat flux.**

We thank the reviewer for catching our mistake. It has been corrected.

**Figure S2B: The vertical axis label "Ice Sheet Velocity" is odd. It's seismic wave velocity, right?**

Yes, we have changed the label to "seismic velocity" as to not confuse it at the speed at which the ice is flowing.

**Figure S4: Please enlarge the study site and consider drawing contour lines.**

**Otherwise, the color scale map does not tell a lot about the hydraulic potential distributions around the lake.**

We prefer to show a broader regional context because the hydraulic potential does not vary perceptibly in the subglacial region since it is dominated by the surface topography of the ice.

**Model description for Greenland Paper**

In order to estimate the temperature in the ice above the lake, we use the steady state conservation of energy:

$$\rho c \frac{\partial T}{\partial t} = 0 = \underbrace{\frac{\partial}{\partial x_i} k_{ij} \frac{\partial T}{\partial x_j}}_{\text{diffusion}} - \underbrace{\rho c \dot{u}_k \cdot \frac{\partial T}{\partial x_k}}_{\text{advection}} - \underbrace{\dot{Q}}_{\text{sources}} \quad (1)$$

where $T$ is the temperature, $\rho$ is density, $c$ is the specific heat capacity, and $\dot{u}$ is the velocity. Tensor indices $i, j, k$ are defined as 1 and 2 being in the horizontal along and across flow directions and 3 as the vertical. The conductivity, $k$. The sources are combined into $\dot{Q}$ and for this case they include both the geothermal flux and that due to latent heat of melting or freezing at the lake ice boundary: $\dot{Q}_{\text{freeze}} = -L\dot{m}$ where $L$ is the latent heat for ice and $\dot{m}$ is the melt rate. Freezing of ice (negative $\dot{n}$) generates heat at the lake interface.

In order to apply this to the ice over the lake, we make several simplifying assumptions:

1. We assume one dimensional geometry. For our low-sloping icefield, this is a reasonable assumption for several reasons. Considering a typical lapse rate of 7°K per kilometer, $\frac{\partial T}{\partial x_1} \sim \frac{\partial T}{\partial x_2} \ll \frac{\partial T}{\partial x_3}$; therefore, even though we have a non-zero horizontal along-flow velocity, the effect of the advection of temperature from upstream is negligible compared to the vertical temperature gradient.

2. We assume that the vertical velocity linearly decreases from the surface (Cuffey and Paterson, 2010)

3. We assume that ice density is constant and equal to 920 kg/m³. This assumption is weak for a compacting firn column, however our firn column is small compared to the full ice depth and we estimate an uncertainty due to this assumption of less than 0.1° C. We could however, we can estimate the effect of differing densities by varying the diffusivity (conductivity and specific heat).

4. We assume the conductivity (2.3 W/m/K) and specific heat (2000 J/kg/K) are uniform. This assumption results in an uncertainty of similarly less than 0.1° C.

5. We assume that the melt or freezing rates at the lake/ice boundary are small enough that the ice thickness is not changing significantly and we can assume steady state.

6. We assume that there is no convection or other currents within the lake and therefore that the bottom boundary condition is the heat flux at lake/ice boundary which is a combination of geothermal flux and melting or freezing.

We vary the surface temperature, the geothermal flux, the freezing rate, and the surface vertical velocity (the accumulation rate in ice equivalent) over a range of values to test hypotheses for lake water temperature.

$$\frac{k}{\rho c} \frac{\partial^2 T}{\partial x_3^2} - \dot{u}_3 \frac{\partial T}{\partial x_3} = \dot{Q}_{\text{geo}} + \dot{Q}_{\text{freeze}} \quad (2)$$

We solve this using a control volume method (e.g. Patankar, 1980).

**Fig. 1.**

[Figure]

[Figure]

**Fig. 2.**

---

## Author Comment (AC3) · 6 Feb 2021

We thank Dr. Buffo for his comments and his interest in our work. Below, we give our replies to each comment.

**1) I believe in the current manuscript the geothermal heat flux labels of Figure 6b are mislabeled and need to be switched.**

Our mistake has been corrected.

**2) I do not feel the 1D thermal model of the ice sheet is described in enough detail so as to reproduce or validate the presented results. There is a broad reference to Patankar (1980) but this text focuses on general numerical methods rather than the setup for the specific ice sheet problem discussed here. What**

is the advection term utilized here? Is it the deposition rate? Accumulation rates are given in 'ice equivalent' form, but are these deposited at the already compacted ice density of 920 kg/m$^3$ or at a lower density and then compacted? I think expanding on the description of the model would help to clarify the utility of the results.

We agree, and have included a thorough description of the thermal modeling to the Supporting Information section of the manuscript. See also the attached document.

**3) At no point are the reflectivity results gathered over the presumed lake (either GPR or seismic) quantitatively compared to the surrounding bedrock reflectivity values. This seems like a missed opportunity to me. The difference in expected reflectivity between bedrock and an ice-water phase transition is discussed, and hypothetical reflection coefficients are plotted in Figure 4, however it is not demonstrated that this is observed in the current study site. I find results comparing such contrasts in reflectivity crucial to the validity of these types of studies - for example Rutishauser et al (2018) "Discovery of a hypersaline subglacial lake complex beneath Devon Ice Cap, Canadian Arctic" present relative power measurements that show striking contrast between regions with lakes and the surrounding bedrock. I feel a comparable approach could be taken in this manuscript to substantially bolster the evidence for the existence of a lake. I do not feel qualitative inspection of the radargram in Figure 2 is enough evidence to conclude that a lake is present. Why are reflection coefficients for regions not directly over the lake excluded from Figure 4 (when this could validate the claims made in the manuscript)? Without an explicit example of contrasting properties between the purported lake and surrounding terrain I do not feel that the conclusion of a substantial (10-15 m thick) lake existing beneath the ice is a valid one.**

This is a good point that requires further clarification. Firstly, the seismic reflection coefficients were excluded from the region beyond the boundary of the lake simply

because it is difficult to make clear amplitude measurements of the R2 arrival in this region, which is necessary to compute the reflection coefficient. It can be seen in Figure 2A that reflection R2 is much more difficult to identify, and at some transect distances (e.g., between roughly 1600 km and 1900 km) seems to almost entirely disappear. In the updated manuscript, we attempt to make measurements of $C_R$ in this region. However, given the very low signal strength of R2, it is not clear whether or not we are simply picking noise. If the results are accurate, it suggests that there is no clear change in the reflection coefficient across the boundary. While we choose not to interpret the seismic reflectivity results in the region beyond the lake boundary, we include the results in the updated Figure 4 (see below), so that the reader can decide for themselves. Additionally, we have added some GPR reflectivity results to the manuscript. We find that the reflectivity is approximately 10 dB larger above the lake, which is in good agreement with Palmer et al., (2013) who found a 10 - 20 dB anomaly associated with the lakes.

The results of Rutishauser et al. (2018) are interesting and relevant, but there is not a strong reason to believe that the basal conditions and materials should be similar in the two field regions. In the Devon ice cap, Rutishauser et al. propose that the hypersaline subglacial lakes are present in bedrock troughs. Hence, a strong contrast in reflection coefficient across between the lake and bedrock is expected. However, in our Greenland field site, there is no conclusive evidence that the region surrounding the subglacial lake is bedrock. Indeed, if the basal material is soft and possibly water-saturated sediment, there should not be a large difference between the seismic reflectivity compared with a subglacial lake.

Finally, we disagree with the statement that our interpretation of the presence of a subglacial lake is based solely on "qualitative inspection of the radargram in Figure 2". In Figure 5, we show the results of detailed seismic modeling which provides evidence for our interpretation, by showing that a thin ( 12 m ) lake satisfies the traveltime and polarities of the seismic observations. Any interpretation should be able to explain

i) A flat reflector with a strong seismic reflection coefficient.

ii) Two strong seismic reflections with opposite polarities (i.e., the phases we interpret as the lake top and lake bottom).

iii) The presence of only one single strong reflection present in the GPR data, which likely indicates that the radar energy is strongly attenuated below the surface of the reflector.

In our opinion, a subglacial lake is the simplest explanation for all of these observations. However, if our assumption of the attenuation in the ice is incorrect, it is possible that we could be over estimating the magnitude of the reflection coefficient. In this case, it is plausible that water saturated dilatant till could explain the reflection amplitudes. In the updated manuscript we clarify that our results are not completely conclusive, although we favor the subglacial lake hypothesis.

**Model description for Greenland Paper**

In order to estimate the temperature in the ice above the lake, we use the steady state conservation of energy:

$$\rho c \frac{\partial T}{\partial t} = 0 = \underbrace{\frac{\partial}{\partial x_i} k_{ij} \frac{\partial T}{\partial x_j}}_{\text{diffusion}} - \underbrace{\rho c \dot{u}_k \cdot \frac{\partial T}{\partial x_k}}_{\text{advection}} - \underbrace{\dot{Q}}_{\text{sources}} \qquad (1)$$

where $T$ is the temperature, $\rho$ is density, $c$ is the specific heat capacity, and $\dot{u}$ is the velocity. Tensor indices $i, j, k$ are defined as 1 and 2 being in the horizontal along and across flow directions and 3 as the vertical. The conductivity, $k$. The sources are combined into $\dot{Q}$ and for this case they include both the geothermal flux and that due to latent heat of melting or freezing at the lake ice boundary: $\dot{Q}_{\text{freeze}} = -L\dot{m}$ where $L$ is the latent heat for ice and $\dot{m}$ is the melt rate. Freezing of ice (negative $\dot{n}$) generates heat at the lake interface.

In order to apply this to the ice over the lake, we make several simplifying assumptions:

1. We assume one dimensional geometry. For our low-sloping icefield, this is a reasonable assumption for several reasons. Considering a typical lapse rate of 7°K per kilometer, $\frac{\partial T}{\partial x_1} \sim \frac{\partial T}{\partial x_2} \ll \frac{\partial T}{\partial x_3}$; therefore, even though we have a non-zero horizontal along-flow velocity, the effect of the advection of temperature from upstream is negligible compared to the vertical temperature gradient.

2. We assume that the vertical velocity linearly decreases from the surface (Cuffey and Paterson, 2010)

3. We assume that ice density is constant and equal to $920\,\text{kg/m}^3$. This assumption is weak for a compacting firn column, however our firn column is small compared to the full ice depth and we estimate an uncertainty due to this assumption of less than 0.1° C. We could however, we can estimate the effect of differing densities by varying the diffusivity (conductivity and specific heat).

4. We assume the conductivity (2.3 W/m/K) and specific heat (2000 J/kg/K) are uniform. This assumption results in an uncertainty of similarly less than 0.1° C.

5. We assume that the melt or freezing rates at the lake/ice boundary are small enough that the ice thickness is not changing significantly and we can assume steady state.

6. We assume that there is no convection or other currents within the lake and therefore that the bottom boundary condition is the heat flux at lake/ice boundary which is a combination of geothermal flux and melting or freezing.

We vary the surface temperature, the geothermal flux, the freezing rate, and the surface vertical velocity (the accumulation rate in ice equivalent) over a range of values to test hypotheses for lake water temperature.

$$\frac{k}{\rho c} \frac{\partial^2 T}{\partial x_3^2} - \dot{u}_3 \frac{\partial T}{\partial x_3} = \dot{Q}_{\text{geo}} + \dot{Q}_{\text{freeze}} \qquad (2)$$

We solve this using a control volume method (e.g. Patankar, 1980).

**Fig. 1.**

[Figure]

**Fig. 2.**

---

## Author Response (AR1)

Dear Editor,

After carefully considering the comments that we received on our manuscript "Geophysical constraints on the properties of a subglacial lake in northwest Greenland" we have modified the manuscript for resubmission. Below, we list each comment (in blue) and give our response (in black).

**Anonymous Referee #1**

This manuscript presents a targeted geophysical study of a Greenland subglacial lake which combines seismic and GPR data to constrain the physical properties of a subglacial lake. The seismic portion of the analysis is largely persuasive and provides a depth measurement for a lake that had previously been detected by airborne radar sounding. However, the GPR portion of the analysis is underdeveloped. Specifically: - The reflectivity of the subglacial lake in the GPR data was not presented or analyzed at a sufficient level of depth. This should be compared and combined with the seismic data to provide a more complete and quantitative picture. - The GPR reflectivity signal is also not sufficiently compared to the airborne radar sounding data in the Palmer paper that motivated the study or to more recent radar studies of Greenland subglacial hydrology and thermal state like: Chu, W., et al. "Complex basal thermal transition near the onset of Petermann Glacier, Greenland." Journal of Geophysical Research: Earth Surface 123.5 (2018) or Jordan, T., et al. "A constraint upon the basal water distribution and thermal state of the Greenland Ice Sheet from radar bed echoes." Cryosphere (2018). - One of the most exciting opportunities from combining seismic and GPR data over lakes is the ability to use the sensitivity of radar attenuation to water conductivity, which the authors mention in the context of EM surveys, to constrain lake salinity/conductivity. The paper would be stronger if this was included.

We agree that the GPR analysis needed improvement. We have added more details to Section 2.1 to clarify the data collection and processing methods. Additionally, we have updated the radar image in Figure 3 which now shows a sharper picture of the ice bottom reflection. Finally, we included some analysis of the ice bottom reflectivity (see last paragraph in Section 3). We find that the reflectivity above the lake is approximately 10 dB stronger than the surrounding region, which is broadly consistent with the results of Palmer et al (2013), who found a variation of between 10 - 20 dB. However, we disagree that a detailed discussion of the differences with Palmer et al. (2013) would be useful. Comparing radar reflectivities obtained from multiple surveys conducted at different times (and here different collection methods) is challenging and only rarely done (e.g., Schroeder, D., Hilger, A., Paden, J., Young, D., & Corr, H. (2018). Ocean access beneath the southwest tributary of Pine Island Glacier, West Antarctica. Annals of Glaciology, 59(76pt1), 10-15. doi:10.1017/aog.2017.45).

Using the radar reflections to constrain salinity would be an exciting possibility. However, in our GPR results, there are no clear returns from signals that have traversed the lake (i.e., lake bottom reflections). This is likely because the water layer is too highly attenuating. The highest

likelihood of detecting lake bottom reflections may be near the edge of the lake where the water layer is thin, yet we can not confidently interpret any signals beyond the primary ice bottom reflection near the lake boundary. We have added some discussion about this to the manuscript.

**Anonymous Referee #2**

1. General comments:
This paper reports seismic and ground-based radar measurements on a subglacial lake in northwestern Greenland. The target of the study is the subglacial lake firstly discovered in Greenland in 2013 based on airborne radar measurements. Some other radar surveys were performed at other lakes beneath the Greenland ice sheet, but this is the first ground-based lake observation in Greenland. The seismic signals enabled the authors to quantify the water depth of the lake, as well as to estimate the material underneath the lake. One dimensional thermal analysis suggested that the lake is filled with hypersaline water under a condition of well below water freezing temperature. In contrast to increasing number of studies on subglacial lakes in Antarctica, much less is reported and known about those in Greenland. The seismic data presented in this paper are valuable, because they provide information below ice-water surface, namely water depth and lake-bed constitution. Similar studies have been performed at some lakes in Antarctica, but this is the first case in Greenland. Numerical analysis of ice temperature is simple, but enough to provide insights into lake water composition and origin of the lake formation. Because of these reasons, I think the reported data are valuable and of great interests of the journal readers. The paper is clearly written. However, it is too concise in some parts and essential information is missing. In general, my impression is that details of method, data and analysis are not sufficiently presented as expected in a paper published in this journal. I am also concerned about the structure of the sections. I list below my concerns, which are followed by more specific comments and corrections. I hope they are considered to improve the paper.

2. Major concerns
(1) Presentation of methodology Some essential information is missing about the measurements and analysis used in this study. For example, radar device is described only by "a 10 MHz monopulse radar system". Information of the manufacturer, type of antenna, receiver-transmitter distance, the way of data acquisition and dragging the device (sledge?) should be described. Another example is ice temperature analysis. Only available information for this computation is "1D steady state advection-diffusion heat transfer model solved using the control volume method". How do you compute vertical strain rate? What is spatial resolution? Any influence of neglected firn layer and horizontal advection? Please describe all these details in the Method section.

We appreciate the suggestion, and have added more detailed description of the GPR data acquisition and processing to the Methods section, and have added a new section to the

Supporting Information that provides a detailed description of the thermal modeling, along with its assumptions (see the new Section 3 in the Supporting Information)

(2) Presentation of data Results of the seismic and radar survey are presented in a limited way (Figs 2, 3 and 5). They are given only by plotting amplitude or power in a grey scale on a time-space domain. I wonder how the authors determine reflections at ice-water and water-bottom reflections. Fig. 2B and Fig 3A show important boundaries, but it gives me an impression that they were drawn only by visual inspection. Further, the authors discuss the phase of the seismic signals to identify the material under the ice. Nevertheless, there is no plot clearly showing such an important observation. I think more details, particularly plots of amplitude/power against time, are necessary to convince the readers of the interpretations and discussions.

The seismic reflections analyzed here (i.e., from either the ice or lake bottom) are clearly distinguishable from other phases based on their moveout; reflected energy arrives at all geophones in the line nearly simultaneously due to the near vertical incidence. In other words, there is no ambiguity of whether or not the phases we are interpreting are subsurface reflections. The reviewer states that it appears that the ice bottom and lake bottom reflections appear to have been drawn based only on visual inspection, yet this is common practice and well accepted. Additionally, we are not certain what is meant by "results of the seismic and radar survey are presented in a limited way", since the cross sections shown in Figures 2A and 3 show all of the relevant data collected from the seismic and GPR surveys, respectively. The presentation style (i.e., using grey scale colormaps) is a very common way of displaying reflection results, and to us, seems to be a matter of personal preference. However, we do agree that the phase of the seismic reflections (particularly the opposite polarity of R1 with respect to the seismic source), was not clearly demonstrated. Thus, we have added a new section to the supplement that outlines the polarity analysis, and includes a new figure that more clearly demonstrates the opposite polarity between the source and the R1 reflection (see the new Section 2 and Figure S4 in the Supporting Information).

(3) Comparison with previous studies Seismic survey on a glacial lake is new in Greenland, but available for lakes beneath the Antarctic ice sheet. Interpretation of the seismic signals should be carried out based on the knowledge obtained in Antarctica. Such studies in Antarctica include those reported in Whillans Ice Stream and Lake Ellsworth. Important previous work exists also in Devon Ice Cap in the Canadian Antarctica. Considering the proximity of the sites and possible similarity in water property, closer comparison of the thermal conditions, geographical and geological settings should be performed. Please also introduce these previous studies more in detail in the Introduction section. I would like to read what are known about water depth, lake-bed constitution, water properties in subglacial lakes in Antarctica and other regions.

We have included a more thorough summary of previous active source seismic surveys conducted in Antarctica, which provides more context for the present study. However, we point out that many of these regions have fundamentally different geological histories and properties,

and there is no strong reason to expect any similarity in subglacial water properties. Therefore, direct comparisons are challenging.

(4) Construction of the sections The paper suffers from mixing of method, results and discussion in the text, particularly in the Methods section. The Methods section begins with study site, and a little of methodology of seismic and GPR measurements (2.1 Field experiment). Then, it explains a bit more about the seismic measurement and directly goes into data and interpretation (2.2 Seismic and GPR imaging). Next subsection explains the analysis of the reflection power, which is followed by interpretation of the data (2.3 Basal reflectivity). This is not usual as a journal article and not convenient for readers. Please consider reconstruction of the text. The best for readers is to explain all the methodology in the Method section, which is followed by presentation of data in detail but without interpretation in the Results section, and finally interpretation and discussion in the Discussion section. I also find the last paragraph of the Introduction section includes too much results and conclusion. I would expect this kind of summary of the study in Abstract, which is currently rather weak.

We agree that a restructuring of the paper was warranted. In the new version the results are clearly separated from the methods in a new section titled Results. Additionally, we have removed the last paragraph of the introduction and moved the main points to the abstract.

3. Specific comments:
Line 15-20: This abstract can be improved by incorporating the essential results of the measurements and conclusion described in the last paragraph of Introduction (Line 68-77).

See response above.

Line 32: "Bentley et al., 2011" » Missing in the reference list (or the publication year is wrong).

The publication year was wrong in the references and we have corrected it.

Line 36-37: "airborne radio-echo sounding" » "airborne" is not a necessary condition. Snow vehicle or snow mobile are also used for surveying lakes.

We deleted the word "airborne".

Line 51: "approximately 40% of ..." » This is not consistent with 124 out of 400 as described in Line 40.

The discrepancy likely comes from different author determinations of "active" subglacial lakes. We have removed the inconsistency, and prefer to reference Smith et al. (2009), who found evidence of 124 active subglacial lakes in Antarctica.

Line 81: Please provide coordinates and elevation of the lake.

We have added a table with the latitudes, longitudes, and elevations of our seismic shot locations to the Supporting Information (see Table S1).

Line 82: Can you indicate the 980 km2 drainage basin on Figure 1B?

We refer the reader to Figure 1 of Palmer et al. (2013), who provide a plot of regional bed topography.

Line 86: "24 40 Hz" » Hyphen is missing.

Fixed.

Line 96: "longitudinal seismic reflection image" » Here and other places, the authors use "longitudinal" and "across", which are not clear to explain settings. Here, for example, "seismic reflection image along the survey route" is better if I understand it correctly.

We agree that "longitudinal" was unnecessary, and have removed it.

Line 107-108: "An additional reflection with opposite polarity of R1" is not clearly shown by Fig. 2B. Also not clear why you think "which is consistent with a lake bottom reflection".

The opposite polarities of R1 (i.e., the "lake top" reflection) and the secondary reflection we identify as "the lake bottom" reflection is more clearly demonstrated in Figure 5A, which shows a close up view of the lake reflection sequence on a single shot gather collected above the lake (shot gather number 12). It is expected that lake top and lake bottom reflections will have opposite polarities because of the opposite reflection coefficients between a layer of ice over water (negative reflection coefficient), and a layer of water over sediment (positive reflection coefficient). This is indeed supported by our modeling results shown in Figure 5B and 5C.

Line 110: What do you mean by "across the seismic section"?

We have replaced "across the seismic section" with "as a function of distance along the transect".

Line 111-112: Uncertainty due to wave velocity is evaluated, but I wonder if there is additional uncertainty due to signal peak determination. How do you define the reflection boundaries in Figure 2B?

This is a good point. There is likely to be variation between different analysts in terms of how they preprocess their data and how they determine phase arrivals and amplitudes, which would introduce some uncertainty. However, this uncertainty is difficult to quantify, and is always prevalent in any such seismic analysis. In this study, we do not believe that the travel time uncertainty would have a large impact on our lake depth results because a difference of several

ms in travel time picks would translate to only small changes in the inferred lake depth (less than 5 m or so). When determining the lake thickness, we prefer to pick the first breaks (of R1 and the subsequent 'lake bottom' reflection) in the processed seismic image.

Line 116: "across the majority of the transect" » "across" is confusing.

Replaced "across" with "along".

Line 117-118: "lake is slightly deeper" » Do you mean "ice is slightly thicker"?

Yes. We have added clarification.

Line 118: Please define "transect distance".

The transect distance is the distance along the profiles shown in Fig 1C. We have added labels "W" "X" "Y" and "Z" to the map in Fig 1C and to our cross sections, which should make this clear.

Line 126: "A_R1 and A_R2" » The variable "A" should be in italic?

We have replaced all instances of A_R1 and A_R2 to be formatted consistently with how they are presented in Equation 1.

Line 160: "IMBIE Team Report" » The author name is inconsistent with the reference list.

Fixed.

Line 160: Can you provide an estimate of "net storage capacity of all of Greenland's subglacial lakes"?

I think that the uncertainties on such an estimate would be too large to make a useful addition. The first-order assumption that would need to be made is that all of the high reflectivity regions identified in airborne radar surveys (e.g., Bowling et al. 2020) represent subglacial lakes with similar depth than the one we identify in this study, which does not seem justifiable.

Line 180: How do you know the surface temperature in the region?

The surface temperature is determined from RACMO2 modeling. This has been made explicit in the text (see lines 86 - 87 in the updated manuscript).

Line 181-182: "the basal temperature ... be well below the pressure dependent melting point" » Why do you think so?

Our estimates of basal temperature are determined through the thermal modeling results summarized in Figure 6.

Line 185: "1D steady state advection-diffusion heat transfer model" » Please describe more details with equations to be solved.

We have added a new section to the Supporting Information that fully describes the thermal modeling, including the equations that are solved and the assumptions that we make.

Line 190: "When advection is ignored" » I understand that you ignore vertical ice motion. It is confusing because you also neglect horizontal advection of ice. Ice flow is small near the divide and down glacier advection of cold ice does not influence the conclusion about basal temperature below melting point, but mentioning the horizontal ice flow helps the readers.

By "when advection is ignored", we are referring to cases in which ice does not accumulate. When accumulation is considered (i.e., when we consider 'advection'), the thermal profiles are altered because the near surface isotherms are moved to deeper depths. We clarified this in the manuscript.

Line 199: Do you have estimate of the salinity from the computed basal temperature? Can you discuss your results with the study at Devon Ice Cap?

Based on our thermal modeling that suggests a basal temperature of between -12 C and -14 C, we estimate that the salinity required to keep the lake liquid would need to be between 160 and 180 ppt. This is comparable to the results of Rutishauser, who suggested a salinity of 140 - 160 ppt for the Devon Ice Cap region. We have added this discussion to the manuscript.

Line 199-200: "ice surrounding the lake would be frozen" » Do you think the hypersaline condition is limited with in the lake area? Such condition may extend to the surrounding area and cause basal melting outside of the lake.

This is a good point and we agree that the hypersaline condition may not be limited to the lake. Discussion has been added to Section 3.1 of the manuscript.

Line 210-219: I agree that continuous supply of surface meltwater to the bed is not likely because meltwater production is limited in this elevation range. Near the study site, a Japanese research group has been running an automatic weather station (e.g. Aoki et la., 2014), performed in-situ snow observations and ice core studies (e.g. Niwano et al., 2015; Kurosaki et al., 2020). I suggest the author to discuss water availability in the region based on the climatic conditions and the previous studies.
- Aoki, T. et al. (2014). Field activities of the "Snow Impurity and Glacial Microbe effects on abrupt warming in the Arctic" (SIGMA) Project in Greenland in 2011-2013. Bulletin of Glaciological Research. 32. 3-20. 10.5331/bgr.32.3. - Niwano, M. et al. (2015). Numerical simulation of extreme snowmelt observed at the SIGMA-A site,

northwest Greenland, during summer 2012. The Cryosphere. 9. 2015. 10.5194/tc9-971-2015. - Kurosaki, Y. et al. (2020). Reconstruction of Sea Ice Concentration in
Northern Baffin Bay Using Deuterium Excess in a Coastal Ice Core From the Northwestern
Greenland Ice Sheet. Journal of Geophysical Research: Atmospheres. 125.
10.1029/2019JD031668.

While it is important to consider water availability, there are no obvious pathways for surface meltwater to recharge the subglacial lake (e.g., moulins / crevasses), even if it were available.

Line 235: "cryoconcentration" » Is this a right word to explain lake formation due to "latent heat from freezing".

We agree this was confusing and have reworded it. The lines now read:
 "Sustaining a freezing rate of several m/yr to generate the latent heat necessary to maintain warm basal ice is less likely than locally elevated geothermal anomaly. We, therefore narrow the lake origin hypotheses to either anomalously high geothermal flux or hypersalinity due to local ancient evaporite."

Line 235-239: It is odd to read this conclusion within the same paragraph explaining "Latent heat from freezing". Please consider to change the paragraph, or merge these sentences with the next paragraph.

Done.

Line 266: "Peters et al., 2013" » Missing in the reference list (or the publication year is wrong).

Thank you for catching this. The correct citation is "Palmer et al. 2013"

Line 272: "hydropotential modeling" » "hydropotential analysis"?

Done.

Figure 1c: Please label the ends of the GPR and seismic survey profiles (e.g. "X" and "Y") so that you can use the labels on Figures 2 and 3.

We have added labels "W" "X" "Y" "Z" to the map in figure 1c and the corresponding cross sections.

Figure 6B: There is something wrong with the line colors. I would expect warmer temperature for the higher geothermal heat flux. Figure S2B: The vertical axis label "Ice Sheet Velocity" is odd. It's seismic wave velocity, right?

We thank the reviewer for catching our mistake. It has been corrected.

Figure S4: Please enlarge the study site and consider drawing contour lines. Otherwise, the color scale map does not tell a lot about the hydraulic potential distributions around the lake.

We prefer to show a broader regional context because the hydraulic potential does not vary perceptibly in the subglacial region since it is dominated by the surface topography of the ice.

**Jacob Buffo**
**jacobbuffo91@gmail.com**

Hello,
First and foremost congratulations and excellent work on a paper which utilizes an array of geophysical methods to probe the existence and properties of a subglacial hydrological feature that has broad implications for a number of scientific fields, including glaciology, climate science, and planetary science. I do however have a number of comments and concerns with the current manuscript:

1) I believe in the current manuscript the geothermal heat flux labels of Figure 6b are mislabeled and need to be switched.

Thank you for catching our mistake.

2) I do not feel the 1D thermal model of the ice sheet is described in enough detail so as to reproduce or validate the presented results. There is a broad reference to Patankar (1980) but this text focuses on general numerical methods rather than the setup for the specific ice sheet problem discussed here. What is the advection term utilized here? Is it the deposition rate? Accumulation rates are given in 'ice equivalent' form, but are these deposited at the already compacted ice density of 920 kg/mˆ3 or at a lower density and then compacted? I think expanding on the description of the model would help to clarify the utility of the results.

We agree, and have included a thorough description of the thermal modeling to the Supporting Information (see Section 3 in SI).

3) At no point are the reflectivity results gathered over the presumed lake (either GPR or seismic) quantitatively compared to the surrounding bedrock reflectivity values. This seems like a missed opportunity to me. The difference in expected reflectivity between bedrock and an ice-water phase transition is discussed, and hypothetical reflection coefficients are plotted in Figure 4, however it is not demonstrated that this is observed in the current study site. I find results comparing such contrasts in reflectivity crucial to the validity of these types of studies - for example Rutishauser et al (2018) "Discovery of a hypersaline subglacial lake complex beneath Devon Ice Cap, Canadian Arctic" present relative power measurements that show striking contrast between regions with lakes and the surrounding bedrock. I feel a comparable approach could be taken in this manuscript to substantially bolster the evidence for the existence of a lake. I do not feel qualitative inspection of the radargram in Figure 2 is enough evidence to conclude that a lake is present. Why are reflection coefficients for regions not

directly over the lake excluded from Figure 4 (when this could validate the claims made in the manuscript)? Without an explicit example of contrasting properties between the purported lake and surrounding terrain I do not feel that the conclusion of a substantial (10-15 m thick) lake existing beneath the ice is a valid one.

This is a good point that requires further clarification. Firstly, the seismic reflection coefficients were excluded from the region beyond the boundary of the lake simply because it is difficult to make clear amplitude measurements of the R2 arrival in this region, which is necessary to compute the reflection coefficient. It can be seen in Figure 2A that reflection R2 is much more difficult to identify, and at some transect distances (e.g., between roughly 1600 km and 1900 km) seems to almost entirely disappear. In the updated manuscript, we attempt to make measurements of $C_R$ in this region. However, given the very low signal strength of R2, it is not clear whether or not we are simply picking noise. If the results are accurate, it suggests that there is no clear change in the reflection coefficient across the boundary. While we choose not to interpret the seismic reflectivity results in the region beyond the lake boundary, we include the results in the updated Figure 4, so that the reader can decide for themselves. Additionally, we have added some GPR reflectivity results to the manuscript. We find that the reflectivity is approximately 10 dB larger above the lake, which is in good agreement with Palmer et al., (2013) who found a 10 - 20 dB anomaly associated with the lakes.

The results of Rutishauser et al. (2018) are interesting and relevant, but there is not a strong reason to believe that the basal conditions and materials should be similar in the two field regions. In the Devon ice cap, Rutishauser et al. propose that the hypersaline subglacial lakes are present in bedrock troughs. Hence, a strong contrast in reflection coefficient across between the lake and bedrock is expected. However, in our Greenland field site, there is no conclusive evidence that the region surrounding the subglacial lake is bedrock. Indeed, if the basal material is soft and possibly water-saturated sediment, there should not be a large difference between the seismic reflectivity compared with a subglacial lake.

Finally, we disagree with the statement that our interpretation of the presence of a subglacial lake is based solely on "qualitative inspection of the radargram in Figure 2". In Figure 5, we show the results of detailed seismic modeling which provides evidence for our interpretation, by showing that a thin (~ 12 m ) lake satisfies the traveltime and polarities of the seismic observations. Any interpretation should be able to explain

      i) A flat reflector with a strong seismic reflection coefficient.

      ii) Two strong seismic reflections with opposite polarities (i.e., the phases we interpret as the lake top and lake bottom).

      iii) The presence of only one single strong reflection present in the GPR data, which likely indicates that the radar energy is strongly attenuated below the surface of the reflector.

In our opinion, a subglacial lake is the simplest explanation for all of these observations. However, if our assumption of the attenuation in the ice is incorrect, it is possible that we could be over estimating the magnitude of the reflection coefficient. In this case, it is plausible that water saturated dilatant till could explain the reflection amplitudes. In the updated manuscript we clarify that our results are not completely conclusive, although we favor the subglacial lake hypothesis.

---

## Author Response (AR2)

**Reviewer 1:**

I was reviewer 1 for the previous version of the manuscript and most of my concerns have been addressed, however two remain:

1) The authors only state that the GPR reflectivity is 10 dB higher than the surrounding area. As a result readers are not able to evaluate their interpretation or analysis. This issue could be addressed if Figure 3 was expanded to include echo power, the calculated reflectivity, and the sensitivity of that calculated reflectivity to the assumed attenuation rate.

We have added a panel to Figure 3 (shown below) that shows the GPR reflectivity at the base of the ice, which shows the contrast between the reflectivity of the lake and the surrounding region. Depending on the assumed attenuation profile, the relative power of the lake bottom reflection is approximately 4 - 8 dB stronger than the reflection from the adjacent bed.

[Figure]

Updated Figure 3.

The description of this analysis is given in our new Section 2.2, which reads:

2.2 Basal radar reflectivity
*"We estimated the relative basal reflectivity of the bed reflector along the track by first correcting for geometric spreading, then correcting for englacial attenuation assuming the englacial attenuation rate is uniform. This assumption of uniform englacial attenuation is common, but not ideal for this situation because horizontal variability in the thermal structure of the ice is not well constrained. We picked the peak power along our bed profile using a semi-automated picking routine, where the user provides the approximate bed picks to guide the automated routine. We assume an englacial average attenuation rate of -15 dB km-1 which is the lower end on the range of values suggested for northwest Greenland by MacGregor et al. (2015), which are based on internal layers and averaged for ice between 25 and 65% of the ice thickness. We chose the lower end based on fitting a linear curve to peak power versus depth for our data set, which suggests attenuation between -12 dB km-1 and -20 dB km-1. This method, described by Jacobel et al. (2009) and further assessed and compared to other methods by Hills et al. (2020), has limitations for our data set because of the 1) limited depth range, 2) limited spatial sampling, 3) scatter in the data due to noise, 4) it relies on the assumption of uniform horizontal attenuation, and 5) it only applies to the depth range of our data; therefore, we only use this estimate as rough proxy for basal material. Because of uncertainties in the attenuation assumptions, we also provide the correction factors for -25 dB km-1 attenuation."*

2) In their response the authors claim that conductivity analysis would not be possible with, however, recent work in this journal (Tulaczyk, S. M. and Foley, N. T.: The role of electrical conductivity in radar wave reflection from glacier beds, The Cryosphere, 14, 4495–4506, https://doi.org/10.5194/tc-14-4495-2020, 2020.) suggests that some analysis from the reflectivity of the lake surface should be possible as well.

We thank the reviewer for their suggestion, and for bringing this paper to our attention. While we have added a citation to this paper, unfortunately we are not able to perform this analysis to constrain conductivity. This is explained below, in paragraph 2 of our updated Results section. Additionally, we have reached out to the first author of this study, and they agree with us that our conclusions are independent of the analysis described in their paper.

*"In addition, we observe that the bed reflected power is approximately 5 dB higher over the lake compared to the surrounding region (Fig. 3C). Similar to the conclusion of Palmer et al., (2013), which was based on airborne radar, we infer this elevated reflectivity to result from an ice/water interface. However, Tulaczyk & Foley (2020) show that subglacial materials with high conductivity can produce similar reflections to an ice/water interface. Additionally, Tulaczyk & Foley (2020) provide a method using information about phase and multiple frequencies to better distinguish among freshwater, brine, or water- or brine-saturated clay. Our available data, however, are at a single frequency and do not retain phase information; therefore, we do not have sufficient information to distinguish between these high conductivity materials based on radar alone. The secondary seismic reflection discussed above suggests that the lake is water of unknown salinity, rather than saturated sediments."*

**Reviewer 2:**

1.  General comments:

The revised manuscript looks good. The structure of the paper is improved. Introduction tells sufficient background of the study. Additional materials given in Supporting Information provides more details of the seismic reflection. The discussion on the lake origin is still not conclusive, but I think it is sufficient as possible interpretations of the important in-situ data. Please consider several minor comments and corrections listed below.

We thank the reviewer for their careful and thorough comments, which have helped improve the manuscript.

2. Specific comments:

Line 19: I think the citation to Palmer et al. (2013) is not necessary here. Guideline of the journal on the website states "Reference citations should not be included in this section, unless urgently required".

The reference has been removed.

Line 33: "ice sheet" >> ice sheets?

Corrected.

Line 90: "Noel et al., 2018" >> Missing in the reference list.

The missing reference has been added.

Line 92: "cm/yr" >> Here and everywhere, "/" and "^{−1}" are mixed up. The journal guideline says "Units must be written exponentially (e.g. W m–2)". I also suggest to stick on MKS unit system.

Per the journal guidelines, we have changed our units to be written exponentially. Also, when appropriate we stick to the MKS system (for example, we changed all instances of cm to m). The only unit that is not MKS is time, which we sometimes use in years instead of seconds.

Line 113: "were stacked" >> Duplicated.

The duplication is removed.

Line 114: Space is missing between 8 and MHz.

It now reads "8 MHz"

Line 132: "Q" should be italic because it is a variable. This is just one example. Please check all variables in the text.

All variables have been made into italics.

Line 141: "(i.e., a wave that has traveled twice….)" >> This should be explained earlier in the text in Line 119.

This was indeed explained earlier. But it is redundant, so we have removed it.

Line 150: "lake velocity" >> This sounds odd. It's seismic wave velocity.

It now reads "seismic velocity of the lake"

Line 166: "a function distance along …" >> a function of the distance along …?

The typo has been corrected.

Line 171: "strongly negative" >> I do not understand why is "strongly". Just "negative"?

"strongly negative" has been replaced with "negative"

Line 183: "IMBIE Team Report" >>Please indicate the author name (not "Report") consistent with the reference list.
We have noticed that this paper is most commonly cited in the literature as "Shepherd et al. (2019)", and have changed our citation accordingly.

Line 186: "shot gather 12" >> Please explain this number "12".

We now explicitly state that this is the 12 shot gather in our seismic line.

Line 206: "lakes below Antarctica" >> lakes below the Antarctic ice sheet?

The suggested change has been made.

Line 223: "existence of liquid water underneath ice" >> Something is missing in the text. Something like, "existence of liquid water underneath ice at temperature below melting point"?

The liquid water must be above the melting temperature, so adding this seems redundant.

Line 227: "If the lake is hypersaline it can…" >> What is "it"? Should be "lakewater"?

Yes. The sentence now reads: "If the lake is hypersaline the lakewater could remain liquid at low temperatures by depressing the freezing temperature. "

Line 228: "depress the freezing temperature …−1.2 …." >> I would expect a positive number after "depress".

Line 229: "roughly 6x that of" >> Please consider to reword "6x".

The word "roughly" has been removed.

Line 270–271: I don't know if these arguments are correct. Even when the lake is maintained by high geothermal flux, basal melting can be small or zero depending on the thermal flux. Moreover, in the accumulation area, ice usually flows downward and vertical strain rate is negative. Are you really able to distinguish the basal processes from measurements above?

This is a valid point. We have replaced the sentence with:

"For a freshwater lake created by high geothermal flux, the basal ice temperature would be near 0o C, vertical velocity would be downward if melting exceeds accumulation"

Figure 1D: Please explain the red triangles in the caption.

The caption has been updated. The red triangles indicate the geophones.

Figure 6: Please consider a larger font for the text in the plots.

We have increased the font size of the labels in Figure 6.

Line 499: "Modeling" >> Modeled?

Corrected.

Table 1: The papers in the caption are missing in the reference list.

The two papers, "Christianson et al. (2014)" and "Peters et al. (2008)" are now in the references. The table incorrectly read "Palmer et al. (2008)" previously, but it has been updated.

---

## Author Response (AR3)

Dear Editor Podolskiy,

We are pleased to hear that you think our manuscript "*Geophysical constraints on the properties of a subglacial lake in northwest Greenland*" worthy of publication in The Cryosphere, and we thank you for your fine attention to detail in the latest round of revisions. Below, we briefly review your comments (in blue), and note the changes that we have made.

1) "Labels of panels must be included with brackets around letters being lower case (e.g. (a), (b), etc.)." Now they are all shown without brackets and in upper case. This applies to all captions and the main text.

All figure labels in the main text and supplementary information have been changed to use lowercase letters with brackets.

2) "Units must be written exponentially" like in the main text, but now Figs. 3c, the insets in 5b & c, and 6 are using "/".

We apologise for overlooking this formatting issue in previous revisions. All units are now written in exponential format, both in figures and in the text.

3) "Capitalization: only the first word is capitalized in headers". Now formatting of axis labels is inconsistent, e.g., Figs. 1-5 have no capitalizing at all or Fig. 3c capitalizes both words (Relative Power).

All section headers and figure axis labels now follow the convention of having the first word capitalized.

[Data and Code availability]

"Authors are required to provide a statement on how their underlying research data can be accessed. This must be placed as the section "Data availability" at the end of the manuscript."

We have added sections that describe the data and code availability. For now, we have written that all data is available by contacting the corresponding author. However, we are working on getting a DOI for our dataset either through the IRIS DMC or using the Digital Repository at the University of Maryland (DRUM). We request that in order to not delay advancing this paper towards publication, that the DOI be added at a later date, such as during the proof correction of the article.

Additionally, we have added a thorough statement on author contributions, roughly following the CREDIT guidelines.

Fig. S2a caption
-> please, indicate the source/reference of the density profile at DYE-3 used in this figure.

The reference Gundestrup & Hansen (1984) has been added.

2. Polarity analysis
Please add a reference to the statement of the first sentence.

The reference Muto et al. (2019) has been added.

(Fig 4) -> (Fig. 4)

The correction has been made.

Fig. S4 caption -
"B and C respectively" -> please, place a comma before 'respectively'.

The correction has been made.

3. Thermal Modeling,
-> Thermal modeling

The correction has been made.

point 3:

"We could however, we can estimate the effect"
-> We could, however, estimate the effect"

The correction has been made.

[Patankar, 1980] is not shown in the reference list.

We have added Patankar (1980) to the references. It is now in both the references of the main text and the supplementary.

4. Hydraulic Head Estimates
-> Hydraulic head estimates

The correction has been made.

In this section, references are made to Fig. S4, but should be to Fig. S5.

The figure references are now correct.

---

## Author Response (AR4)

Dear Editor Podolskiy,

We apologize for overlooking the attachment with your comments and specific suggested changes. In the latest submitted version, we believe that we have addressed the remaining issues. Below we give a summary of the changes that have been made based on the comments in the attachment. Please let us know if we overlooked any of the issues, and we thank you again for your attention to detail.

Line 100: "After data was collected" was changed to "After data were collected"

Line 110: "km hr$^{-1}$" was changed to "km h$^{-1}$"

Line 121: We added two references which use an assumption of uniform englacial attenuation.

Line 125: We clarified how MacGregor et al. (2015) calculated the englacial attenuation from radar reflections, and added an additional citation.

Line 193: "due to for example" was changed to "due to, for example,"

Line 220: The variable $V_P$ was changed to italics for consistency.

Line 246: We changed "depress the freezing temperature of water by -12° C to -14° C" to "depress the freezing temperature of water by 12° C to 14° C"

Line 269: a comma was added after e.g.

Line 285. "We, therefore" -> "We, therefore,"

Line 297: The comma was removed from Bell et al., (2014)

Figure 4A: The variable $C_R$ was changed to $c_R$ in the x-axis label.

Figure 4B: The font size was increased in the legend for easier readability.

Figure 6 caption: There was a mistake in the description of Fig. 6B. All of the profiles shown have a surface temperature of -22$^{\circ}$C, and it is the heat flux and accumulation rate that are changed. The updated caption has been corrected.

Data availability: We have now added a link to access the active source seismic data, which is stored on the Digital Repository at the University of Maryland (DRUM).

Author contributions: There was a mistake in the initials of one of the coauthors. "AG" was changed to "AM".